# Involvement of G-quadruplex regions in mammalian replication origin activity

Paulina Prorok[1], Marie Artufel[2,8], Antoine Aze[1,8], Philippe Coulombe[1,8], Isabelle Peiffer[1], Laurent Lacroix[3], Aurore Guédin[4], Jean-Louis Mergny [4,5], Julia Damaschke[6], Aloys Schepers[6,7], Benoit Ballester [2] & Marcel Méchali[1]

Genome-wide studies of DNA replication origins revealed that origins preferentially associate with an Origin G-rich Repeated Element (OGRE), potentially forming G-quadruplexes (G4). Here, we functionally address their requirements for DNA replication initiation in a series of independent approaches. Deletion of the OGRE/G4 sequence strongly decreased the corresponding origin activity. Conversely, the insertion of an OGRE/G4 element created a new replication origin. This element also promoted replication of episomal EBV vectors lacking the viral origin, but not if the OGRE/G4 sequence was deleted. A potent G4 ligand, PhenDC3, stabilized G4s but did not alter the global origin activity. However, a set of new, G4-associated origins was created, whereas suppressed origins were largely G4-free. In vitro *Xenopus laevis* replication systems showed that OGRE/G4 sequences are involved in the activation of DNA replication, but not in the pre-replication complex formation. Altogether, these results converge to the functional importance of OGRE/G4 elements in DNA replication initiation.

[1] Institute of Human Genetics, CNRS-University of Montpellier, 141 rue de la Cardonille, 34396 Montpellier, France. [2] Aix Marseille Univ, INSERM, TAGC, Marseille, France. [3] Balasubramanian group, Department of Chemistry, University of Cambridge, Lensfield Road, Cambridge CB2 1EW, UK. [4] ARNA Laboratory, Université de Bordeaux, Inserm U1212, CNRS UMR5320, Institut Européen de Chimie Biologie (IECB), Pessac 33607, France. [5] Institut Curie, CNRS UMR9187, Inserm U1196, Universite Paris Saclay, Orsay, France. [6] Research Unit Gene Vectors, Helmholtz Zentrum München (GmbH), German Research Center for Environmental Health, Marchioninistraße 25, 81377 Munich, Germany. [7] Monoclonal Antibody Core Facility & Research Group, Institute for Diabetes and Obesity, Helmholtz Zentrum München, Ingolstädter Landstrasse, 85764 Neuherberg, Germany. [8] These authors contributed equally: Marie Artufel, Antoine Aze, Philippe Coulombe. Correspondence and requests for materials should be addressed to B.B. (email: benoit.ballester@inserm.fr) or to M.M. (email: marcel.mechali@igh.cnrs.fr)

In mammals, around 100,000 potential DNA replication origins (origins throughout the text) are distributed along chromosomes. However, only about 30% is activated in a cell, in an apparent stochastic way. This flexibility in origin choice is considered an important feature for the robustness of DNA replication, and for the adaptation to DNA replication stress and cell fates (for a review)[1]. The second main feature of metazoan origins is their sequence plasticity. Indeed, differently from *Saccharomyces cerevisiae* origins, metazoan origins do not have a unique conserved consensus element. Some genetic and epigenetic characteristics have been identified in the vicinity of origins, but none can be considered to be a universal feature of metazoan origins. Among these features, the Origin G-rich Repeated Element (OGRE) is present in more than 60% of origins, in fly, mouse, and human cells[2–6]. This element can potentially form a G quadruplex (G4) structure (thereafter, such sequence elements are defined as OGRE/G4), and it is upstream of the initiation site (IS) of DNA synthesis, at an average distance of 250–300 bp. This localization could be compatible with the position of the pre-replication complex (pre-RC), and is associated with a nucleosome-free region[4]. The presence of similar elements at human origins has been detected using a different method than those used for mouse[2,6] and chicken cells[7], and it was shown that proteins involved in DNA synthesis initiation, ORC[8], MTBP[9] and MCM2–7[10] are also associated to such elements. A functional evidence for the use of this element was reported in chicken cells in a 1.1 kb fragment of the β-globin replication origin flanked by an HS4 insulator included close to a blasticidin resistance transgene under the control of the strong actin promoter[7]. However, it is unclear whether this result can be translated to other model systems, and no analysis has been done so far on a natural replication origin, at its original site or at an ectopic position.

Here we used various experimental approaches to determine whether OGRE/G4 is a functional element at metazoan origins. First, using an in vivo genetic approach at an endogenous locus, we showed that deletion of this motif strongly reduced origin activity in mouse cells. Moreover, an OGRE/G4-containig sequence introduced in an ectopic origin-free region promoted the establishment of a new functional origin. Second, we showed that a plasmid containing an origin with an OGRE/G4 element can replicate in HEK293 cells that express EBNA1 almost as efficiently as plasmids containing the Epstein-Barr virus (EBV) origin OriP, and that deletion of the OGRE/G4 element strongly reduces its replication efficiency. Third, we analyzed the influence of PhenDC3, a known G4 ligand, on origin firing efficiency genome-wide. Fourth, we performed competition experiments in in vitro systems of DNA replication derived from *Xenopus laevis* eggs, and found that G4-forming sequences are competitors that strongly affect DNA replication initiation.

Altogether, all our results converge to the conclusion that G-rich elements, including the OGRE/G4 motif, are functionally important for origin activity.

## Results

**OGRE/G4 elements can form G4 in vitro.** We first asked whether the OGRE/G4 motif could form G4 in vitro. Origins were identified from which cells by purification of **S**hort RNA-primed **N**ascent **S**trands (SNS), a procedure that we and others repeatedly found to be accurate for origin analysis in *Drosophila melanogaster*[5], mouse[2–4] *Arabidopsis thaliana*[11], *Caenorhabditis elegans*[12], chicken[7], and human cells[13–15], and the results of which were confirmed by different approaches[6,10,14,15]. Supplementary Fig. 1 summarizes this procedure (detailed in "Methods" section), and shows the controls used for this analysis.

We tested the capacity of G4 formation by sequences found in the origin vicinity using isothermal difference spectra (IDS) and circular dichroism (CD). To test their propensity to form a G4-structure, we selected origins in different chromatin domains, transcription status and replication activity. Because each sequence needed to be individually synthesized and tested by CD and IDS, we did a selection of 7 origins. The bioinformatics prediction for a potential of G4-structure was first tested at the bioinformatical level, using the G4H algorithm (similar results were obtained with the Quadparser software), and indicated a high capacity for G4-formation for all tested sequenced (Fig. 1a and Supplementary Table 1). Circular dichroism (CD) is a highly sensitive assay, which can determine the conformational state of quadruplex structures[16]. Isothermal differential spectra (IDS) are obtained using a method derived from that for thermal denaturation spectra[17]; they provide information on the nature of the folded structure. Both assays showed that all these sequences exhibited hallmarks of quadruplex formation, as shown by the strong negative peak around 295 nm and the two positive peaks around 240 and 273 nm for IDS (Fig. 1a, left panel), and the strong positive peak around 260 nm with CD (Fig. 1a, right panel). Such data suggested a predominantly parallel quadruplex conformation for all sequences and confirmed G4 formation by these sequences.

**OGRE/G4 elements confer replication origin activity.** We then selected a strong and reproducible origin that was present in all our five independent experimental replicates (Ori 1, Supplementary Table 1; Supplementary Fig. 2A shows the raw data in our replicates). The replication origin positions were defined in a genome-wide manner using MACS2 and SICER peaks calling softwares, as previously described[4]. The origin initiation site is the highest NS-enrichment score over the initiation region. The OGRE/G4 motif was located 240 nt upstream of the IS (Fig. 1b and Supplementary Fig. 2A), in agreement with previous results in mouse cells[4]. After insertion of a 1907 bp fragment that included the OGRE/G4-containing Ori1 into a large region devoid of replication or transcription activity (Fig. 1b, Supplementary Fig. 2A–C and "Methods" section), we tested replication activity by SNS purification followed by qPCR with primers for the inserted origin sequence (Supplementary Table 2 and "Methods" section). The replication profiles showed that Ori1 was active at the ectopic position (Fig. 1c). As the inserted sequence was identical to the original sequence, the origin activity observed after the insertion was around twice the activity measured in parental cells. Conversely, the activity of another origin on chromosome 11 (external origin, Ori2) did not change (Fig. 1c).

To functionally assess the importance of the OGRE/G4 motif, we also used another experimental system based on the replication of episomal DNA in mammalian cells. This episomal plasmid harbors the EBV origin OriP that is recognized by the viral protein EBNA1[18]. OriP is a bipartite element consisting of the family of repeats (FR) and the dyad symmetry (DS) element. Both are recognized by EBNA1, favoring the mitotic segregation of the episome and DNA replication during S phase respectively[19]. Interestingly, replication occurs ORC dependently once per cell cycle in synchrony with chromosome replication[20–22].

After transient transfection of different episomal plasmids (Fig. 1d, left panel) in HEK293 cells that stably express EBNA1, we analyzed episomal DNA replication by DpnI digestion/ transformation (Fig. 1d and "Methods" section). DS deletion (deltaOriP) strongly inhibited episomal DNA replication, showing the requirement of an active origin in this system (Fig. 1d,

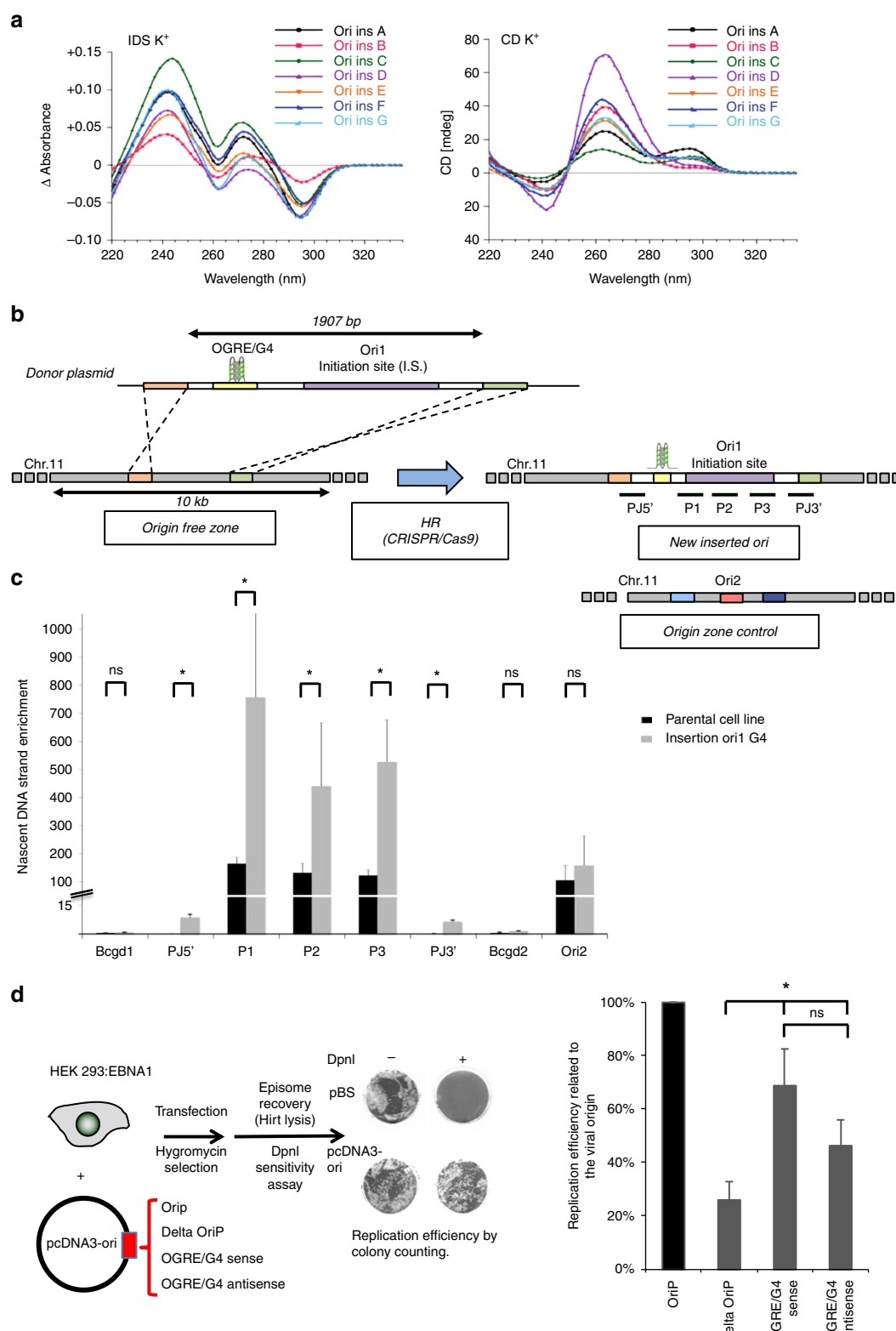

right panel). Insertion of a 500 bp mouse OGRE/G4-containing origin (Ori2; Supplementary Table 1) at the place of OriP (Fig. 1d) also promoted episomal DNA replication almost as efficiently as the viral origin. As previously shown, the OGRE/G4 presence is orientated relative to the initiation site, as initiation occurs always downstream to the OGRE/G4[3,4]. So, when the antisense sequence is used, the initiation site will be in the other

direction. In this orientation the origin is still active, as expected, although slightly less possibly because of a different chromatin environment in the reverse direction.

From these results, obtained in two different in vivo systems and using different methods to analyze origin activity, we concluded that an OGRE/G4-containing origin can function ectopically in the genome and also in episomal plasmids.

**Fig. 1** Creation of an ectopic DNA replication origin. **a** Isothermal differential spectra (IDS; left panel) and circular dichroism spectra (CD; right panel) of potential OGRE/G4 sequences fond in the vicinity of replication origins. All tested sequences form G4 structures, as indicated by the strong negative peak around 295 nm and the two positive peaks around 240 and 273 nm (for IDS), and the strong positive peak around 260 nm (CD). The sequences are provided in Supplementary Table 1. **b** Ori1 that contains an OGRE/G4 element 240 bp upstream of the DNA replication initiation site (IS) was inserted by Cas9-stimulated homologous recombination into an origin-free region on chromosome 11 in NIH 3T3 mouse cells. The insertion of the 1907 bp fragment (marked in violet) occurred thanks to the two 500 bp homology arms (orange and green) present on the insertion template. The position of the primers (P) (sequences in Supplementary Table 2) used for the analysis of origin activity is also shown. **c** Ori1 activity in parental (control in black) and recombinant NIH 3T3 cells (in grey). As expected, a two-fold increase in DNA replication activity was detected in recombinant cells compared with parental cells, whereas the external origin Ori2 exhibited the same replication activity in both cell lines. Note that SNS activity was also detected at the 5′ and 3′ junctions of the insertion site, but not in the corresponding control regions. The background control regions Bcgd1 and 2 are located in origin-free regions; results are the mean ± SD of 3 independent experiments; *p* values were obtained using the two-tailed Student's *t* test; *$p \leq 0.05$, $p > 0.05$. **d** Analysis of DNA replication, using the DpnI digestion method and colony counting ("Methods" section), in an EBV episomal plasmid transfected in HEK293 cells that express EBNA1. DNA replication activity was assayed using EBV episomal plasmids that carry or not (Delta) the OriP origin, or a 500 bp fragment of Ori2 containing an OGRE/G4 element in the sense or antisense orientation. Results are the mean ± SD of 3–7 independent experiments; *p* values were obtained using the two-tailed Student's *t* test; *$p \leq 0.05$; ns not significant, $p > 0.05$

**Deletion of the OGRE/G4 inhibits replication origin activity**. To further confirm that the potential formation of a G4 is important for the origin functionality, we deleted the endogenous OGRE/G4 sequence in Ori1. Co-expression of the Cas9 nickase and two gRNAs targeting this sequence led to the formation of a double-strand break and the subsequent deletion of the targeted sequence (Fig. 2a, "Methods" section and Supplementary Fig. 3A). The strong peak (G4H score > 2) observed in the wild type sequence with G4-Hunter (a tool to predict the propensity of a sequence to form G4) disappeared for both mutated alleles (no signal above 1), strongly suggesting that our targeted deletion removed the putative G4-forming sequence at this locus (Fig. 2b). In order to confirm these predictions, we analysed the circular dichroism (CD) (Supplementary Fig. 3B, left panel) and isothermal differential spectra (IDS) (Supplementary Fig. 3B, right panel) of Ori1 wt sequence and 2 mutated alleles of Ori1. The results indicated a strong capacity of G4-formation by the wt sequence with a strong positive peak around 260 nm on CD spectrum, and a strong negative peak around 295 nm and the two typical positive peaks around 240 and 273 nm on IDS spectrum. In agreement with the bioinformatics predictions these hallmarks of G4-formation are lost in mutated Ori1 alleles. It is noteworthy that the bioinformatics predictions gave a very accurate prediction of G4-forming potential that was confirmed by in vitro CD and IDS spectra analysis for all tested sequences (Fig. 1a, Supplementary Fig. 3B, C).

Quantification of the origin activity by SNS purification and qPCR analysis showed that in mutant cells, Ori1 replication activity was decreased by 85%, but not that of an external origin (Ori2), also located on chromosome 11 (Fig. 2c). The transcription levels of the *Rai1* gene, associated with Ori1, and of the *Actb* (actin) and *Gapdh* controls were only slightly affected (Fig. 2d), making unlikely an indirect effect due to a transcriptional activity change.

Similarly, deletion of the OGRE/G4 sequence in the episomal vector strongly inhibited episomal DNA replication (Fig. 2e). Randomization of the OGRE/G4 sequence also decreased origin efficiency, suggesting that, at least for Ori2, G-richness *per se* is not sufficient and that G4 formation is an important feature (Fig. 2e). Additionally, to confim the capacity of G4-formation by Ori2 and its absence in Randomised Ori2 we analysed the circular dichroism (CD) (Supplementary Fig. 3C, left panel) and isothermal differential spectra (IDS) (Supplementary Fig. 3C, right panel) of Ori2 wt sequence and Randomised Ori2. The results unambiguously showed a strong G4-forming potential in the wt sequence that was completely lost in Random mutant.

Altogether, these functional studies indicate that the OGRE/G4 element located upstream of Ori1 is functionally active and positively contributes to origin activity.

**G4-stabilization increases G4-associated origins firing**. To better understand the importance of OGRE/G4 elements, we investigated genome-wide whether G4 stabilization could affect origin activity in mouse embryonic stem (ES) cells. We used PhenDC3 (Fig. 3a), a bisquinolinium compound that has high affinity for G4 and that shows an exceptional selectivity for G-quadruplexes[23,24] compared with duplexes, as indicated by the increase in melting temperature ($\Delta T_{1/2}$; stabilization) of seven different quadruplexes, but not for the control duplex (FdxT) (Supplementary Fig. 4A, and Supplementary Table 3).

Compared with control ES cells, incubation with 10 μM PhenDC3 for 48 h, as previously described[25], did not affect the cell cycle profile (Supplementary Fig. 4B), and the expression and phosphorylation of CHK1, a kinase involved in cell cycle progression and in the DNA damage checkpoint (Supplementary Fig. 4C). Conversely, CHK1 phosphorylation was induced by the genotoxic agents camptothecin (Cpt) and etoposide (Eto). The expression of OCT4, a pluripotency marker, also was not modified by PhenDC3 (Supplementary Fig. 4C).

We used a Volcano plot to identify statistically significant changes in replication origin activity (Fig. 3b). A Volcano plot visualizes the biological effect on the *x*-axis (Log2(fold change, FC)) and the statistical significance on the y-axis ($-\log10$(false discovery rate, FDR)). This analysis allowed to define five origin classes according to their activity in response to PhenDC3: *insensitive*, *new*, *reinforced*, *reduced* or *suppressed* (Fig. 3b–e and Supplementary Table 4). Examples of origins belonging to these classes are shown in Fig. 3c, while the fold change in origin activity for each class is depicted in Supplementary Fig. 4D. Overall, we did not observe a substantial increase of origins in the presence of PhenDC3 (Supplementary Table 4). The heatmap (Fig. 3d) showing read density in the vicinity (±7 kb) of origins indicated that *reduced* and *suppressed* origins were situated in an origin-dense environment as opposed to *reinforced* and *new* origins. Origins that remained at the same position and with a similar activity (PhenDC3 *insensitive*) represented 77.9% of all origins (Fig. 3e). One possible hypothesis could be that formation of a G4 is not essential for the activity of most origins, but this is in contradiction with our functional analyses showing the requirement of the OGRE/G4 element for origin activity (Figs. 1 and 2). A second possibility is that most G4 were normally formed during origin assembly or activation with no need of further stabilization by PhenDC3. It was nevertheless plausible that the genetic, chromatin and transcriptional landscape also influence the activity of G4 origins (see later).

Incubation with PhenDC3 also led to a set of *new* origins (15.7% of all origins) with a level of activity comparable to that of *insensitive* origins (Fig. 3e). Two smaller origin classes were

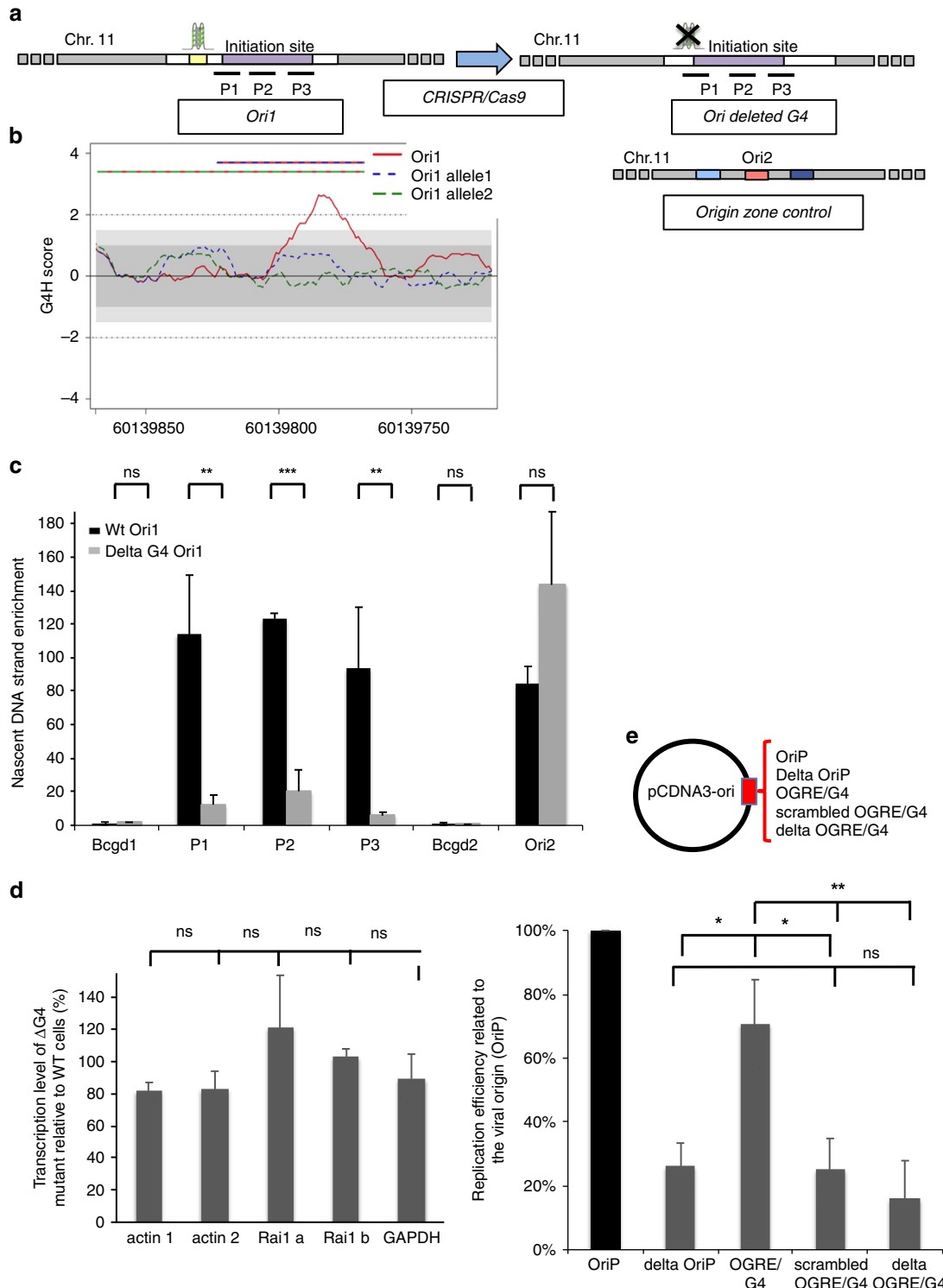

represented by *reinforced* and *reduced* origins (0.6 and 0.7%, respectively). Origins *reduced* by PhenDC3 were initially particularly strong, among the top 10% of strongest origins (Fig. 3e). Overall, we observed that the ligand-mediated G4 stabilization led to a more uniform activity of all origins (Fig. 3e).

We used the RSAT peak-motifs program (see "Methods" section) to find a specific motif in these origin classes. This led to the de novo identification of a G-rich motif upstream the IS, similar to the OGRE/G4 element[2,4], in all origin classes but for

the *suppressed* class (Fig. 4a and Supplementary Fig. 5A). *Suppressed* origins were G4-poor, and preferentially contained a GC-rich motif (Supplementary Fig. 5A), possibility reflecting the enrichment of these origins in GCI promoters and a link with transcription (see below). In *reduced* origins, the OGRE/G4 element was at almost 400 bp upstream of the IS (Fig. 4b). This could be a consequence of their localization close to a promoter. We concluded that G4 stabilization by PhenDC3 did not reveal any new motif in the replication origin repertoire, but led to the

**Fig. 2** OGRE/G4 deletion strongly decreases the DNA replication activity of an endogenous origin. **a** The OGRE/G4 sequence of an endogenous origin (Ori1) was deleted and the deletion was confirmed using a restriction site close to the targeted sequence (see "Methods" section). **b** G4 formation propensity profiling of the Ori1 sequence targeted for deletion. The Ori1 sequence is located on chromosome 11 and presents a strong peak in the G4-Hunter score profile (red line). Such peak is not present upon OGRE/G4 deletion (alleles 1 and 2, blue and green dotted lines, respectively), and no point above 1 or below −1 is observed. This argues against the probability of G4 formation at this mutated locus. The striped lines on the top indicate the extent of the deletion in allele 1 (red and blue) and allele 2 (green and blue). **c** Nascent strand enrichment of Ori1 in parental NIH 3T3 cell line (black) and in mutant clones with the deletion (grey). Replication activity was strongly decreased after deletion of the OGRE/G4 sequence, whereas the activity of the external origin (Ori 2) did not vary. The background control regions Bcgd1 and 2 are located in origin-free regions. Results are the mean ± SD of 3 independent experiments; $p$ values were obtained using the two-tailed Student's $t$ test; *$p \leq 0.05$; **$p \leq 0.01$; ***$p \leq 0.001$, $p > 0.05$. **d** Deletion of the OGRE/G4 did not affect the transcription level of the Rai1 gene, associated with Ori1. As a control, the housekeeping genes Actb and Gapdh were used. Results are the mean ± SD of 3 independent experiments; $p$ values were obtained using the two-tailed Student's $t$ test; $p > 0.05$. Primer sequences are in Supplementary Table 2. **e** DNA replication activity was assessed as in Fig. 1c with the EBV origin, or with the 500 bp OGRE/G4 element of Ori2, or after scrambling or deletion of the same OGRE/G4 sequence. Results are the mean ± SD of 4–5 independent experiments; $p$ values were obtained using the two-tailed Student's $t$ test; *$p \leq 0.05$; **$p \leq 0.01$; $p > 0.05$. Note that data presented for episome containing delta oriP and OGRE/G4-containing origin were performed independently from results presented in the Fig. 1; ns not significant: $p > 0.05$

suppression of a discrete origin population that lack the OGRE/G4 sequence. These origins were in origin-dense regions, and their suppression might compensate the appearance of new OGRE/G4-containing origins, favored by their PhenDC3-mediated stabilization.

We experimentally tested G4 formation in vitro in a subset of *new* origins using CD and IDS, as previously described (Fig. 1a and Supplementary Table 1 for the full list of tested sequences). All these sequences exhibited the hallmarks of quadruplex formation (Fig. 4c). The presence of a minor peak around 295 nm may indicated the formation of alternative folds (possibly anti-parallel G4 structures) for some sequences.

We then asked whether the five origin classes correlated with putative G4 predicted by the G4-Hunter (G4H)[26] and Quadparser (QP) algorithms[27] (Fig. 4d and Supplementary Table 4). The bioinformatics analysis gave highly accurate predictions of G4-forming potential that was confirmed by CD and IDS analysis for several wt and mutant sequences (Fig. 1a, Supplementary Fig. 3B, C). Using stringent parameters, 490,971 G4 were predicted by Quadparser (G-track size min = 3; parameters loop size min = 1, max = 7, Gs permitted in the loop), and 568,806 by G4-Hunter (threshold = 2, window size = 25). Analysis of G4 distribution in each origin class gave similar results with both software programs. This analysis showed no difference in G4 score distribution among classes (Supplementary Fig. 5B; for simplicity, only the results with G4-Hunter are shown), which indicates that G4 strength does not explain our observations.

Moreover, we did not find any significant correlation between the length of the OGRE/G4 sequence and the different origin classes (Supplementary Fig. 5A, C), but we detected a slight global effect of the number of OGRE/G4 motifs present close to the IS (Supplementary Fig. 5D).

Finally, to confirm the functional link between PhenDC3 effect and the OGRE/G4 motifs, we used an indirect FRET melting competition assays with OGRE/G4 oligonucleotide sequences from the *insensitive* class (which were the same as tested for G4-formation using CD et IDS) and *new* origin classes as well as unlabeled positive (G4) and negative controls (single- or double-strand oligonucleotides) (Supplementary Table 1). These sequences were added to a mixture containing a double-fluorescently labeled G4 forming sequence (F21T) corresponding to the human telomeric motif, in the presence or absence of PhenDC3. PhenDC3 bound to F21T and increased its melting temperature in a concentration-dependent manner ($\Delta$Tm = +29 °C at 1 µM and ≈ +18 °C at 0.5 µM; Fig. 4e) when no competitor was present. Negative control competitors, unable to bind to PhenDC3 (dT30 and DS26; single- and double-strands, respectively) did not affect this stabilization, as expected given the

high specificity of PhenDC3 for G4 structures. Conversely, the strong decrease in stabilization observed after addition of origin sequences confirmed the recognition by PhenDC3 of the OGRE/G4 motifs in these origins that acted as strong competitors for PhenDC3 (Fig. 4e). We concluded that PhenDC3 displays high affinity for both *insensitive* and *new* origins, confirming the functional link between PhenDC3 incubation and the observed changes in replication activity.

**Transcription and not G4 govern replication at promoters**. Analysis of the genomic location showed that overall, origins were enriched at gene regions, compared with intergenic regions (Fig. 5a), as previously widely observed[28] and references herein). Remarkably, *suppressed* and *reduced* origins were highly enriched at promoter regions, whereas the other origin classes were mostly absent from promoters and evenly distributed between transcribed and intragenic regions (Fig. 5b, random origins as dotted lines, and control in Supplementary Fig. 6A), confirming our previous results. Next, we asked whether the five origin classes defined in this study were associated with specific chromatin signatures. Pearson correlation analysis using BEDTools[29] (see "Methods" section) revealed that *suppressed* and *reduced* origins were strongly correlated with chromatin marks associated with active transcription and with bivalent epigenetic marks (Fig. 5c, control randomized regions in Supplementary Fig. 6B, and reference data in Supplementary Table 5). *Suppressed* and *reduced* origins were also associated with several transcription factors, further confirming the promoter location of these origins. This result also explains why origins from the reduced class exhibited stronger replication activity before G4-stabilisation. These origins were quite strong because of the presence of both G4 and active transcription. The decreased transcription activity at these origins upon G4 stabilisation decreased the stimulating effect of transcription of these origins. *Reinforced* origins were only slightly correlated with enhancer marks (Fig. 5c). Moreover, we observed a strong link between the formation of *new* origins and regions poor in epigenetic marks, but enriched in G4-forming fully methylated sequences (Fig. 5c).

To further interpret these results, we analyzed the transcriptional output associated with each origin class by RNA-seq analysis of control and PhenDC3-treated ES cell samples ("Methods" section) followed by identification of genes that were differentially expressed in each class using the DESeq2 algorithm (see "Methods" section). We computed the enrichment set using the genes associated with each origin class and by considering the origin localization at the promoter (TSS ± 2 kb, left panel) or within the transcribed regions (TSS + 10 kb). We found that origin activity tended to follow the

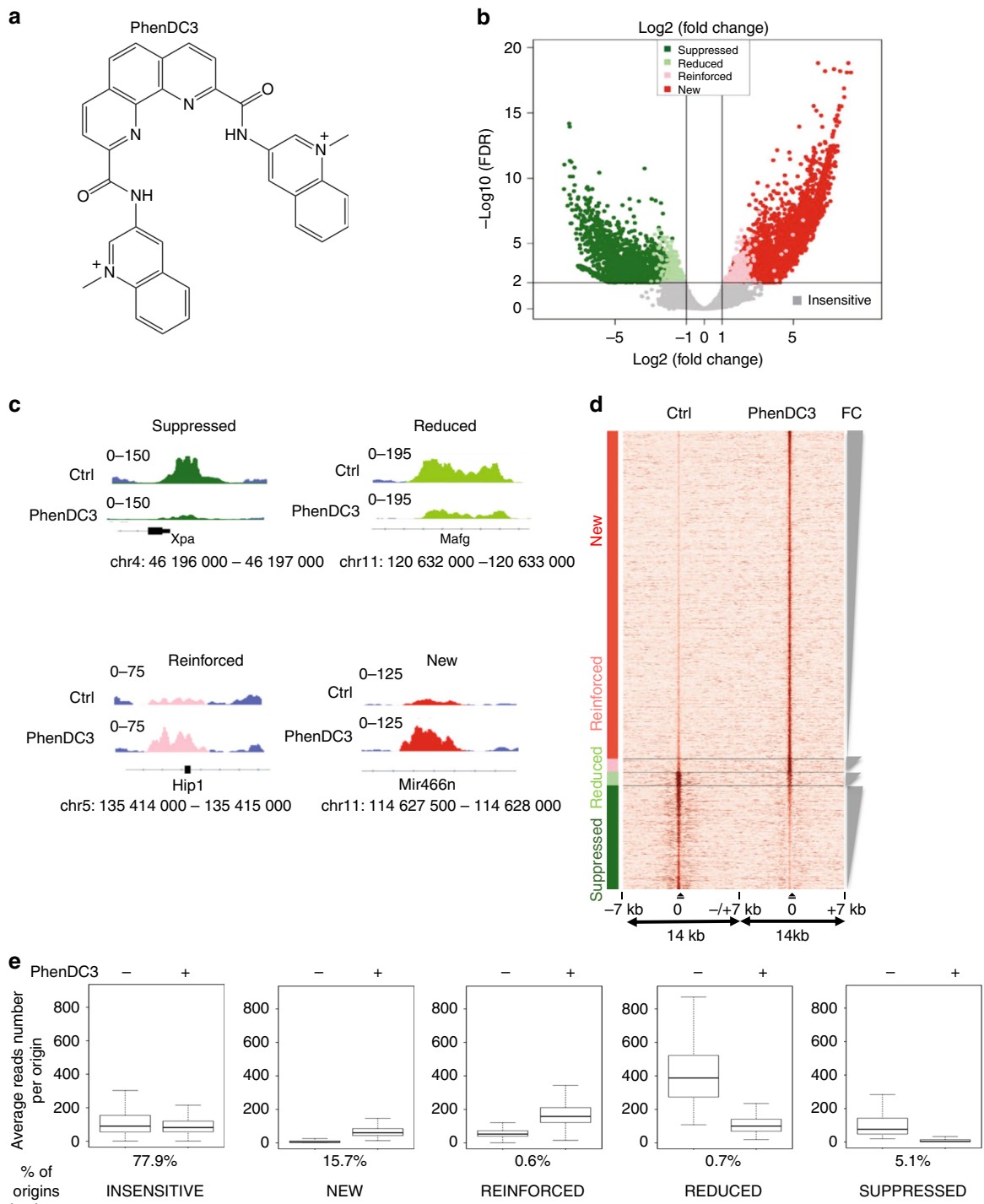

**Fig. 3** Changes in the replication origin repertoire upon G4 stabilization by PhenDC3. **a** PhenDC3 formula. **b** Volcano plot of origins affected by incubation of mouse ES cells with PhenDC3. After identification of the bound sites in all SNS-seq samples, differential binding analysis was performed. For each origin, the corrected *p* values (false discovery rates, −log10(FDR)) and the log2 fold change (FC) of control and PhenDC3-treated samples were plotted. The horizontal and vertical lines correspond to the thresholds for detecting differential origins. On the basis of the FC and peak reproducibility, origins were classified in five different classes, according to PhenDC3 effect (*suppressed, reduced, reinforced, new, and insensitive*), as described in "Methods" section. **c** Examples of the activity of origins in the indicated classes after incubation with PhenDC3 or in control cells. The corresponding genomic region is indicated and the origin color is according to the corresponding class in the Volcano plot. **d** Heatmap showing the read densities in origins affected by G4 stabilization (PhenDC3-treated vs. Control). The heatmap indicates the signal strength (number of reads) and density around each origin and was performed on 7 kb regions on each side of origins, as previously described[4]. The intensity (brown) is proportional to the read counts per 100 bp bins. Origins were sorted on the basis of the FC in signal strength. **e** Activity of origins (reads number) in each class in control (−) and PhenDC3-treated (+) cells

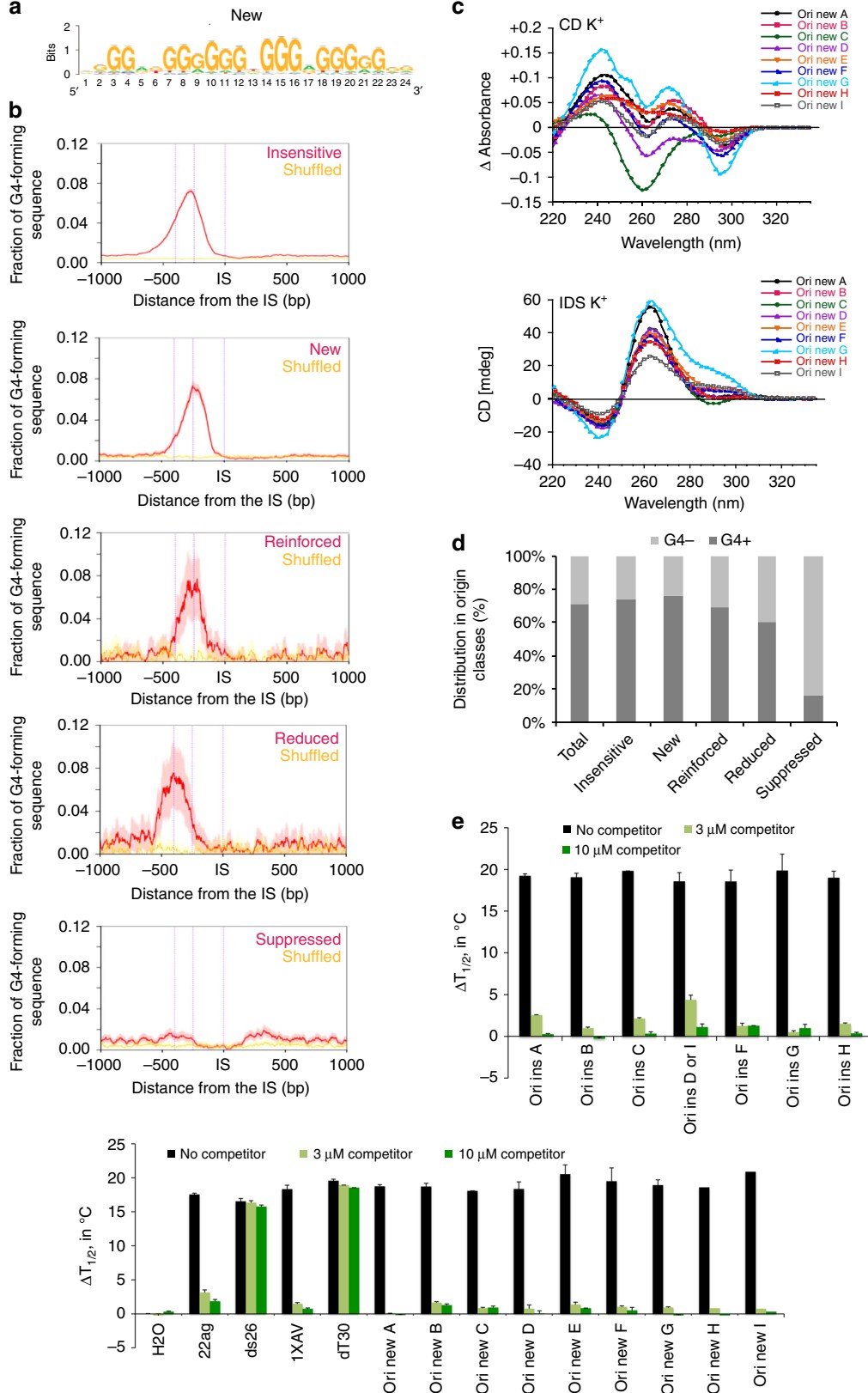

transcriptional output. For instance, *suppressed* origins found at promoters were significantly associated with gene repression (Fig. 5d), as well as *reduced* origins. Conversely, *new* and *reinforced* origins found at promoters tended to be associated with upregulated genes. This is also in agreement with the observation that origins close to TSS are usually highly active, and are downregulated when transcription decreases[1,28,30]. We did not detect any correlation between replication and

**Fig. 4** Nature of the OGRE/G4 in the different origin classes. **a** De novo motif found as the most representative in the new origin class using the RSAT suite[60]. For motifs found in the other classes see Supplementary Fig. 5A. **b** Fraction of OGRE/G4 sequences in function of the distance from the IS. The OGRE/G4 motif forms a relatively sharp peak upstream the IS at an average distance of 250 bp in all origin classes, but for the suppressed class. The CL95% is shown in pink. The fraction of OGRE/G4 sequences in shuffled regions and their CL 95% is shown in yellow and light yellow, respectively. **c** Isothermal differential spectra (IDS; upper panel) and circular dichroism spectra (CD; lower panel) of potential OGRE/G4 sequences associated with the new class of replication origins. All tested sequences form G4 structures, as indicated by the strong negative peak around 295 nm and the two positive peaks around 240 and 273 nm (for IDS, top panel), and the strong positive peak around 260 nm by CD (bottom panel). The IDS suggest that the tested sequences form predominantly G4 in parallel conformation. The possibility to adopt alternative folds, such as anti-parallel G4 structures, for some sequences is indicated by a minor peak around 295 nm. The sequences are provided in Supplementary Table 1. **d** Association of origins with OGRE/G4 motifs in the different classes. Insensitive, new, enforced and reduced origins are mainly G4-associated, but not suppressed origins. **e** FRET competition assays in which stabilization (ΔT1/2, in °C) of the human telomeric quadruplex F21T by 0.5 μM PhenDC3 was analyzed in the absence (black bars), or in the presence of G-rich sequences from insensitive (upper panel) and new (lower panel) origins (3 or 10 μM strand concentration; dark green and light green bars, respectively), of positive (22Ag, 1XAV, both forming G4 structures), and negative (ds26 and dT30 are double- and single-stranded controls, respectively) control sequences. The means were obtained in independent expreiments ± SD. Efficient competition by quadruplex-forming oligonucleotides is evidenced by a sharp drop in stabilization. The origin oligonucleotide sequences are provided in Supplementary Table 1

transcription changes for origins situated in transcribed regions (Fig. 5d, right panel).

We concluded that *i)* replication origins are enriched in transcribed regions, including promoter; and *ii)* origins situated at promoters are often devoid of OGRE/G4 sequences, and their firing activity strongly depends on the transcription level. Conversely, G4 stabilization might facilitate origin firing in non-genic regions that are less prone to chromatin opening, or spontaneous G4-formation, such as fully methylated regions. In these regions, OGRE/G4 might help replication origin activity through its two main features: the presence of single-stranded DNA in the strand opposite to the G4, and its ability to exclude nucleosomes, and to favor a less energetically demanding origin activity in transcriptionally silent regions.

**G4-forming oligonucleotides compete for replication factors**. Initiation of DNA replication is a two-step process. First (i.e., replication licensing), pre-RCs are assembled at origins and this includes the binding of ORC, CDC6, CDT1 and the MCM helicase. Then, the MCM helicase is activated and allows the recruitment of the DNA polymerase machinery. To determine whether OGRE/G4 elements could be potential binding sites for proteins involved in these steps, we performed classical oligonucleotide competition experiments in *Xenopus laevis* low-speed egg extracts (LSE). Xenopus LSE is a well defined cell-free system that faithfully reproduce DNA replication in vitro[31]. This reaction is entirely transcription-independent, thus excluding any influence by the transcription process on the assay. Oligonucleotides similar to the endogenous target DNA sequence should compete for the replication activity as opposed to oligonucleotides which are not related to the target sequence. To test whether OGRE/G4 oligonucleotide templates compete for factors involved in DNA synthesis on sperm nuclear chromatin (Fig. 6a), we incubated *X. laevis* LSEs with 80-mer oligonucleotides that contained an OGRE/G4 sequence (from Ori1 used in the CRISPR/Cas9 experiments), or a sequence with the same G content but randomized (random oligonucleotide), or an AT-rich sequence (Supplementary Table 6), or water (mock), or sonicated salmon sperm DNA. The kinetics of nuclear DNA replication (oligonucleotides do not replicate in the extract) were comparable in mock-treated extracts and after addition of sonicated salmon sperm DNA. DNA replication was slightly delayed by incubation with random and AT-rich oligonucleotides, whereas it was nearly abolished by OGRE/G4 oligonucleotides (Fig. 6a, and quantification in Fig. 6b). Differently from LSEs, *X. laevis* high-speed egg extracts (HSE), in which nuclear membranes have been removed, cannot initiate dsDNA replication. However, they can perform all the reactions occurring

during complementary DNA strand synthesis, as tested with ssM13 DNA as template[32] including RNA priming, elongation and ligation of Okazaki fragments, and chromatin assembly coupled to DNA synthesis. In these extracts, DNA synthesis was not affected by pre-incubation with OGRE/G4 or random oligonucleotides (Fig. 6c). We concluded that OGRE/G4 oligonucleotides compete specifically with replication initiation, and have little or no effect on the subsequent steps.

**G4 are involved in replication origin firing step**. It is unlikely that OGRE/G4 oligonucleotides inhibit DNA replication through checkpoint activation because the DNA damage checkpoint is deficient in *X. laevis* early embryos[33,34]. In agreement, OGRE/G4 oligonucleotides did not induce CHK1 phosphorylation in our in vitro conditions (Supplementary Fig. 7A), differently from incubation with pApT at a concentration that mimics post-midblastula transition conditions known to induce the checkpoint, while pCpG do not[35] (Supplementary Fig. 7A, lane 5). Moreover, caffeine, a checkpoint inhibitor, did not rescue the inhibition of DNA replication by OGRE/G4 oligonucleotides (Supplementary Fig. 7B), whereas it did it in a control experiment where DNA replication was inhibited by aphidicolin (Supplementary Fig. 7C). Altogether, these findings show that checkpoint activation does not explain the inhibition of DNA replication by OGRE/G4 oligonucleotides.

We then investigated which replication initiation step was inhibited by exogenous G4 oligonucleotides. Pre-RC formation can be analyzed in *X. laevis* HSEs that allow this reaction, but not DNA synthesis initiation. Factors involved in origin recognition (ORC5), the recruitment of the MCM helicase onto DNA, (CDC6), and the MCM complex (MCM4) were similarly loaded on chromatin in mock-treated HSEs and in samples incubated with salmon sperm DNA, random oligonucleotides, or OGRE/G4 oligonucleotides (Fig. 6d). Formation of the nuclear membrane also was not affected, as shown by the chromatin recruitment of ELYS, a protein required for the formation of a functional nuclear membrane[35,36] (Fig. 6e). Conversely, the recruitment of CDC45, which is needed for DNA synthesis activation[37], and of factors required for DNA synthesis initiation and for DNA strand elongation (RPA, and PCNA) was strongly decreased (Fig. 6e–f). These results suggest that OGRE/G4 oligonucleotides do not disturb the licensing step of DNA replication, but rather affect the conversion of the pre-RC into the DNA synthesis elongation complex. This result is in agreement with the recent finding that origin firing activity by Mdm2-binding protein (MTBP) in *X. laevis* and human cells is dependent on its G4-binding motif[9].

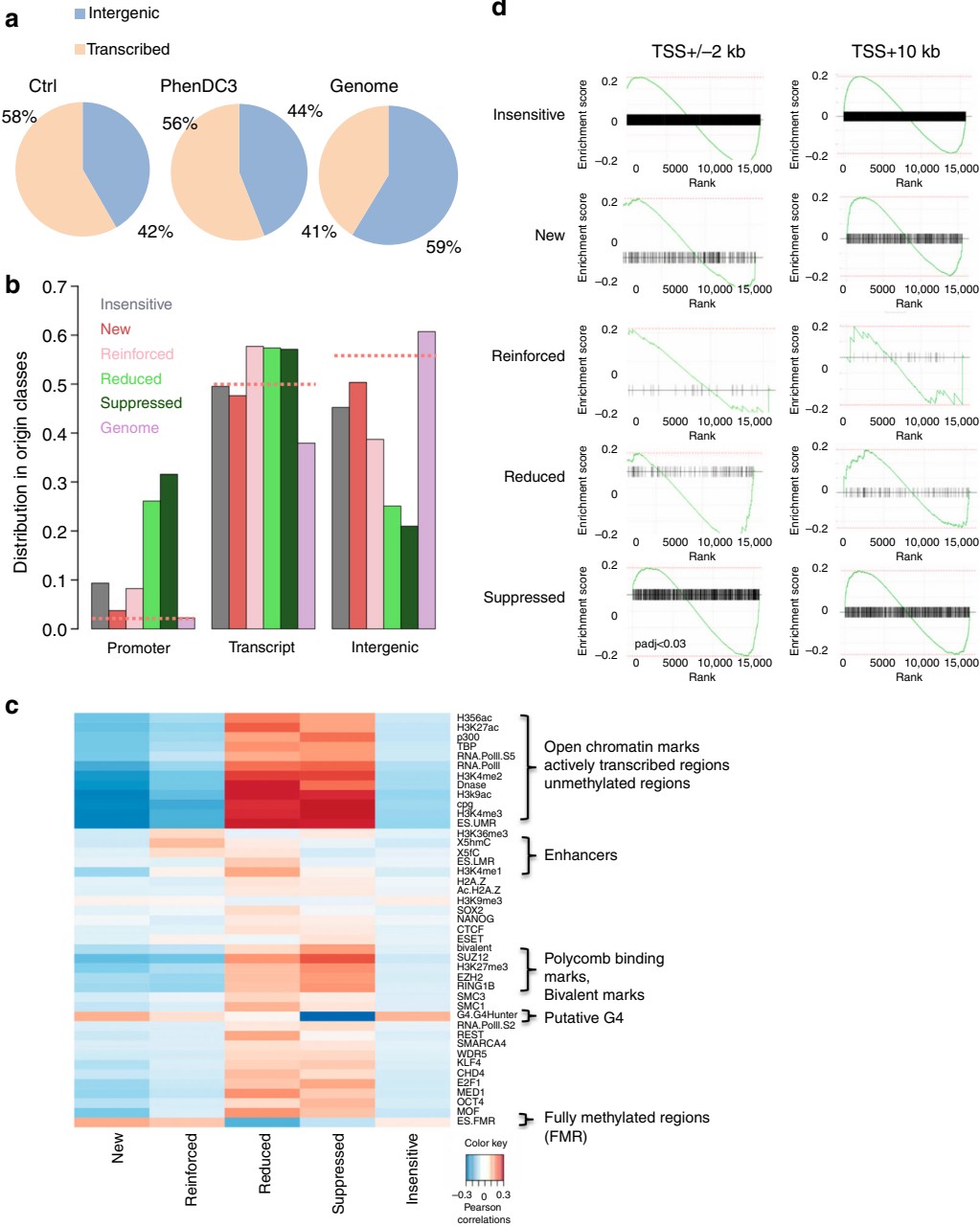

**Fig. 5** Transcription and epigenetic landscape in the different origin classes. **a** Venn diagrams showing the origin distribution between transcribed and intergenic regions in Control and PhenDC3-treated mES cells, and random distribution (Genome). Replication initiation sites are enriched in gene regions. **b** Genomic localization of the different classes of origins relative to transcription. Downregulated origins (suppressed and reduced) are mainly located at promoters. Random origins (dotted lines) are equally distributed in transcription-related regions. **c** Epigenetic marks associated with the different origin classes. All tested open chromatin marks were enriched around reduced and suppressed origins. New and reinforced origins were located mainly in highly methylated regions. **d** GSEA analysis of origins situated at promoters (TSS ± 2 kb, left panels) or in transcribed regions (TSS + 10 kb, right panels) for each class. A plot is drawn for each gene set. The x-axis of each plot represents differentially expressed genes ranked from upregulated (on the left) to downregulated (on the right). The enrichment score is indicated on the y-axis. The black horizontal bar indicates the genes present in the gene set. The highest enrichment score indicates the enrichment. If this score is on the left, the enrichment is higher for upregulated genes; if it is on the right, the enrichment is higher for the downregulated genes. Origins associated with the TSS follow the transcription changes upon G4 stabilization, whereas origins located in transcribed regions are insensitive to changes in transcription levels. At gene promoters, GSEA results show a significant association of downregulated genes with the suppressed origin class (after multi-testing correction using the Benjamini–Hochberg method, adjusted $p < 0.03$). The other origin classes show a similar trend, without reaching significance

## Discussion

Genome-wide analyses of replication initiation profiles first highlighted that metazoan origins were enriched near CpG islands[2,28,38,39]. Then, the G-rich OGRE motif that could potentially form G4 was identified in the mouse and fly genomes[3,4] and subsequently also in mouse[4], chicken[7], fly[5], and human cells[14,40]. This element was discovered using the SNS purification system coupled with high-throughput sequencing (SNS-seq), which has currently the best resolution to map replication origins[41]. Moreover, G4 presence was detected also using λ

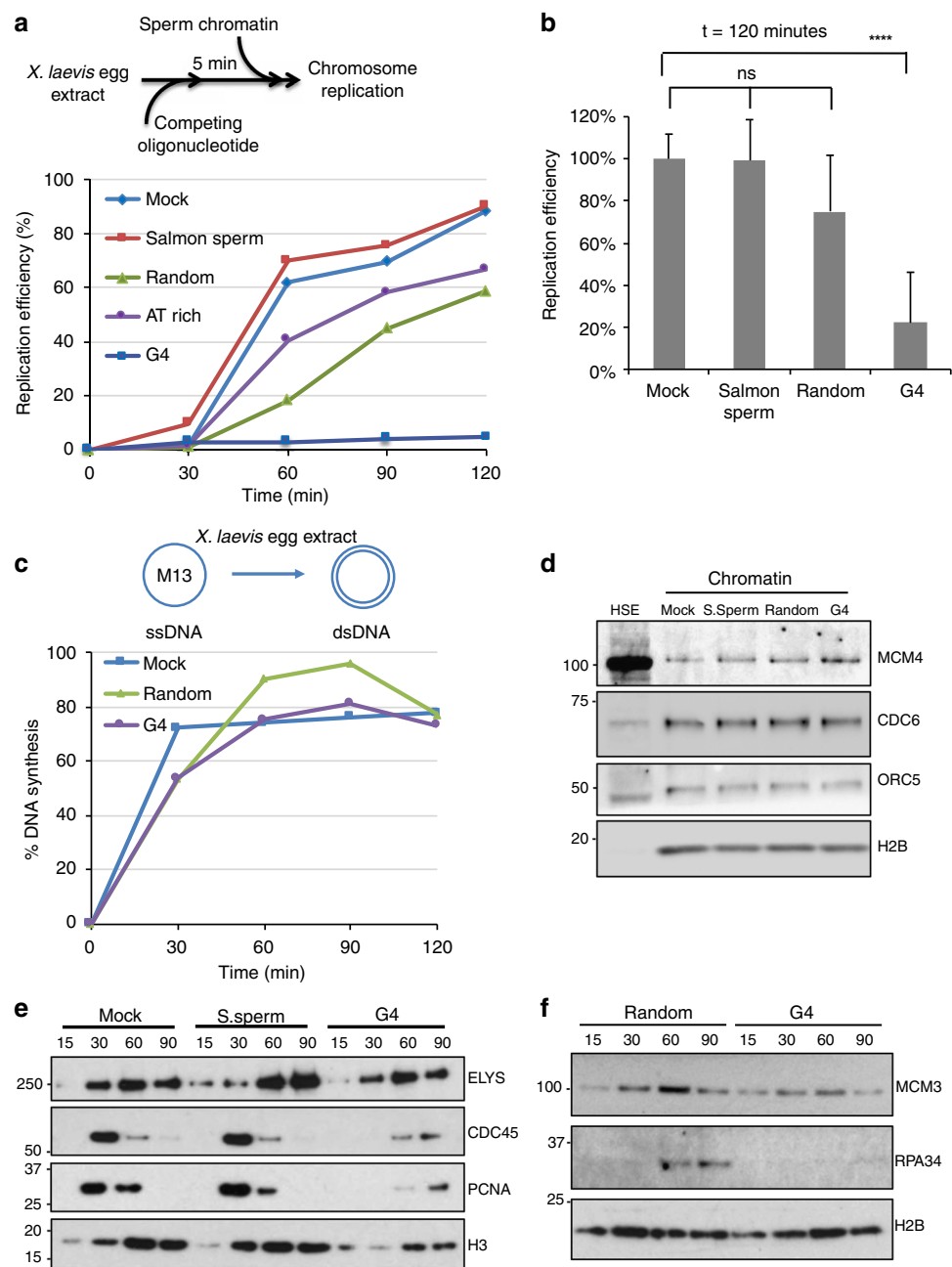

**Fig. 6** At the activation step of DNA replication, OGRE/G4 elements compete for dsDNA but not for ssDNA replication. **a** Schematic representation of the replication kinetics of sperm nuclei in *X. laevis* low-speed egg extracts (LSE) in the presence of competing double-stranded (ds) oligonucleotides. LSEs were pre-incubated with competing oligonucleotides, sonicated salmon sperm DNA, or ultrapure H2O (mock) at 22 °C for 5 min. **b** Average DNA replication efficiency (mean + SD) of LSEs pre-incubated with competing oligonucleotides or controls ($n = 6$ for mock/salmon sperm DNA pre-incubated extracts, $n = 3$ for mock/random oligonucleotides). Total incubation time was 2 h. *P* values were obtained using the two-tailed Student's *t* test; ****$p < 0.0001$, $p > 0.05$. **c** Replication kinetics of ssM13 complementary DNA strand synthesis in *X. laevis* high-speed egg extracts (HSE) pre-incubated with the indicated competing oligonucleotides or H2O (mock). **d** Competition by OGRE/G4 oligonucleotides does not affect pre-RC formation. Sperm nuclei were added to HSEs incubated with H2O (mock), sonicated salmon sperm DNA (S.Sperm), or the indicated oligonucleotides. Chromatin was isolated and immunoblotted with the indicated antibodies. Histone H2B level was used as loading control. **e**, **f** Competition by OGRE/G4 oligonucleotides affects DNA replication activation. Time-course analysis of replication initiation factor recruitment to chromatin after addition of sperm nuclei to LSEs pre-incubated with H2O (mock), sonicated salmon sperm DNA (S. sperm), or competing oligonucleotides. At the indicated time points, chromatin was isolated and immunoblotted with the indicated antibodies; ns not significant

exonuclease-independent conditions[6,15,40], and by genome-wide profiling of human replication origins after pulse labeling of SNS (Ini-Seq)[6,14].

We used several complementary approaches to address the involvement of G-rich repeated elements and their potential to

form G4 structures in the activity of DNA replication origins. Our data confirm that such sequence elements are associated with the majority of active origins, and are localized just upstream of the initiation site. In vivo deletion or insertion of an OGRE/G4-containing wild type origin showed that the OGRE/G4 motif is

functionally active. This result was obtained using origins that are present in the mouse genome, as well as using recombinant episomal DNA.

In our ectopic assay, an OGRE/G4-containing fragment from an origin inserted in a region completely devoid of both DNA replication, transcription activity and G4-forming sequences led to the creation of a functional origin. Deletion of the OGRE/G4 element strongly decreased the activity of the origin. However, we cannot rule out that in other genomic regions, other features might stimulate or repress origin activity. Finally, we found that transcription activity of the gene associated with the origin remained unchanged upon origin deletion, indicating that the link between origin activity and transcription activity is not functionally compulsory. Moreover, our G4-stabilization assay suggests that this link is mostly limited to promoter regions.

OGRE/G4 elements exclude nucleosomes at mouse replication origins[4]. Nucleosome-free regions were also observed in *S. cerevisiae* origins[42–45], although an AT-rich element characterizes their consensus origin-specific ARS element and also plays a role of nucleosome exclusion. OGRE/G4 elements might have a similar function in metazoans. Another possibility is that this sequence is the binding site for a replication initiation factor. In agreement with this hypothesis, recombinant ORC preferentially binds to G4-containing oligonucleotides[8], as well as MTBP, partner of Treslin, that is involved in activation of origins of replication[9]. RIF1, a protein that regulates the timing of origin activation, also binds to putative G4-forming sequences[46]. Putative G4-forming sequences have also been observed at viral replication origins, such as the Kaposi sarcoma associated virus (KSHV) origin. This origin contains several G4 sequences and allows the stable maintenance of the viral episome in cells, and associates with ORC and MCM proteins[47]. Putative G4-forming sequences are also present at the EBV replication origin, to which EBNA1, the viral protein involved in origin recognition, binds[48].

We used PhenDC3 as a G4-binding tool to reveal new G4-related features linked to replication origin activity. Incubation with PhenDC3 did not affect the activity of most origins, despite the presence of putative G4 sequences, suggesting that most origins do not need further stabilization by PhenDC3 for their activity. However, G4 stabilization increased the predisposition to become a replication origin for a subset of OGRE/G4-containing origins. These origins are mainly localized in non-coding regions that are poor in epigenetic marks and enriched in fully methylated regions. We propose that PhenDC3 might facilitate the formation of G4 structures in fully methylated regions that are less favorable to their formation[49,50]. The influence of DNA methylation status on G4-folding capacities has been very recently provided[51] using a G4-recognizing antibody which detected folded G4 structures in hypomethylated regions that overlap with DNMT1 binding sites. DNMT1 is a DNA methyl transferase that restores the DNA methylation pattern just after DNA replication. It has affinity for G4 structures, but surprisingly these structures inhibit its catalytic activity. In this way, DNMT1 can be concentrated in the vicinity of replication start sites and could immediately act on newly synthetized DNA after origin activation. Alternatively, PhenDC3 might facilitate the formation of G4 in heterochromatin structures, and therefore facilitate nucleosome exclusion and the formation of replication initiation complexes.

PhenDC3 incubation also led to the suppression of some origins that lack the OGRE/G4 element. These origins are found in promoters and are significantly associated with gene repression. We suggest that the replication activity of *suppressed* origins is mainly guided by transcription, and is not OGRE/G4-dependent. Transcription increases the activity of all origins close to a TSS when gene transcription is upregulated and decreases their

activity when transcription is downregulated. DNA replication can benefit from the open chromatin structure at gene promoters. However, the influence of transcription activity was limited to promoters, and transcription changes did not affect the activity of origins localized in gene bodies. This is in agreement with the observation that transcriptional silencing of the X chromosome does not induce changes in the strength or localization of the tested origins situated in gene bodies[52]. Finally, PhenDC3 incubation reduced the efficiency of a small fraction of origins (0.7%). These few origins were among the strongest ones in control cells, and were mostly associated with promoters. It is possible that the appearance of new origins upon incubation with PhenDC3 reduced the need of very strong origins.

*New* origins represented 71% of all origins affected by PhenDC3-mediated G4 stabilization, and showed a level of activity similar to that of *insensitive* origins. The appearance of these new OGRE/G4-containing origins might compensate the suppression of origins that lack OGRE/G4.

Examination of specific loci during *X. laevis* early development has shown that initiation of DNA replication did not require specific sites[53,54], in contrast with late development, when site-specific initiation of DNA replication correlates with transcription onset in the embryo[55]. This regulation was explained by the huge excess of replication factors in *X. laevis* eggs, and by the short cell cycle (30 min) without G1 and G2 phases during the first 12 cell cycles after fertilization. Here, we found that OGRE/G4 oligonucleotides, but not random or AT-rich oligonucleotides, are strong competitors for replication origin activity in this system. We showed that this competition is at the level of DNA replication initiation and not at the level of complementary DNA strand synthesis. The pre-incubation with OGRE/G4 oligonucleotides did not affect pre-RC formation on origins, but only DNA synthesis activation. This suggests that some factors involved in this process are sequestered by the competing OGRE/G4 oligonucleotides. Our results might suggest a new explanation to the rapid replication cycles of Xenopus early embryos. Indeed, it is now recognized that potential origins are in large excess relative to those effectively activated in a given cell. The inter-origin spacing in a somatic cell is around 100 kb. If all origins were to be activated in a given cell, this spacing would be less than 10 kb. A full usage of specific origins would be therefore compatible with the speed of DNA replication in *X. laevis* early development.

How could G4 structures be involved in DNA replication initiation? From *E. coli* to higher eukaryotes, origins usually contain an origin recognition site, where the pre-RC is assembled, upstream of the initiation site of DNA synthesis, where nascent DNA strands are initiated by the DNA polymerase machinery. The origin recognition site may play a regulatory role, similar to transcription promoters that are localized 50 to 300 bp upstream of the TSS. An important feature of the OGRE/G4 element is its localization not at the initiation site of DNA synthesis, but 250 bp upstream of it[2–4], suggesting an interaction with factors involved in the pre-RC. Figure 7 illustrate this position and show that our present data also confirm this position. This localization would fit with the site of assembly of the preRC, in agreement with the observation that recombinant ORC preferentially binds to G4 sequences[8]. However, alternatively OGRE/G4 elements could be part of sequences that regulate DNA synthesis initiation, possibly explaining the present discrepancy between its role in origin recognition and its replication fork stalling activity[56]. It is worth noting that our oligonucleotide experiments in Xenopus egg extracts point out to a role in the activation of DNA replication origins rather than in the assembly of the preRC. Known factors involved in this activation step are the kinase activity (DDK) which phosphorylates MCMs subunits and a complex

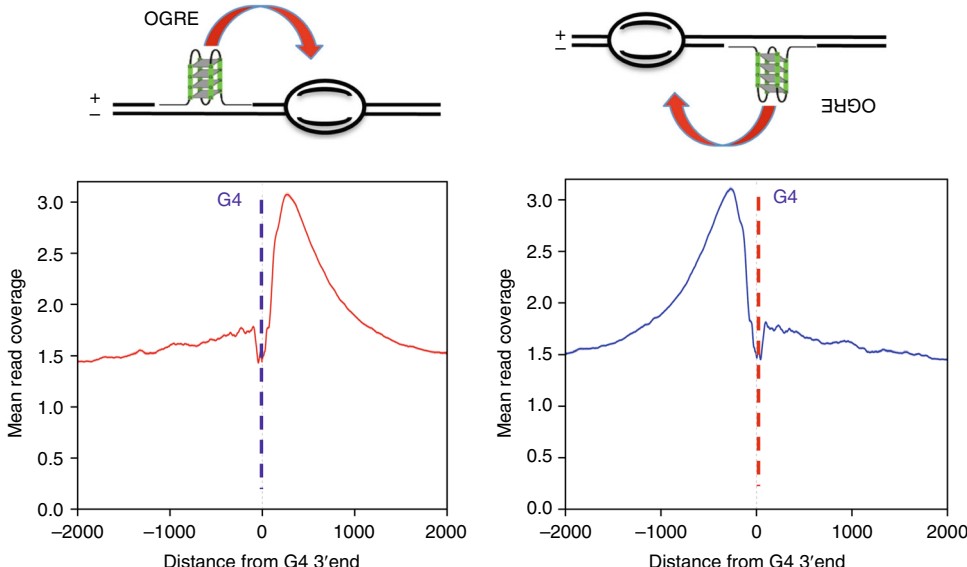

**Fig. 7** G4 function in the DNA replication initiation. Based of the position of the OGRE/G4 that we already reported[2] and the corresponding model[3], we know that the OGRE/G4 element is always upstream of the initiation site itself, either on the + or the − DNA strand (left and right upper panels). In the corresponding lower panels, based on our present data, we confirm again the position of the G4 element relative to the initiation site. The position of the G4 might fit with the position of the preRC, but this does not obligatory imply it will recruit some factors to the pre-RC. First, G4 can adopt several types of structure which themselves might regulate the folding or the replication origin region. Second, G4 might play a role in the removal of the nucleosome positioned at the initiation site itself, a process necessary to load the replication machinery, and therefore regulate activation of DNA replication. Third, because G4 are themselves nucleosome-free regions, they could facilitate DNA helix opening at the initiation site

reaction engaging sevral activating factors factors such as Sld2, Sld3 and GINS, cdc 45, Dbp11. OGRE/G4 elements could play a role in this activation step by helping the recruitment of these factors. G4 sequences can adopt several different G4 structures, which possibly may have different roles in the processing of the preRC to the activation step. Another important feature of OGRE/G4 element is that they are nucleosome-free, in contrast to the initiation site itself which contains a positioned nucleosome[4]. In such, OGRE/G4 may also recruit factors involved in the removal of the positioned nucleosome during the activation step of DNA replication, therefore facilitating the recruitment of the DNA polymerase machinery and its associated factors. From an evolutionary point of view, the use of structural elements, such as a G4-forming sequence, to set the replication program might be advantageous because it is not dependent on strict sequence specificity. As these elements are widely present in the genome, their function could be less affected by potential point mutations than strict consensus sequences.

## Methods

**Cell culture**. CGR8 mouse ES cells cells (obtained from Austin Smith's laboratory, Department of Biochemistry University of Cambridge, UK) were cultured on gelatin-coated dishes (feeder-free, to avoid DNA contamination by mouse embryonic fibroblasts) in Glasgow Minimum Essential Medium (GMEM) supplemented with 2 mM glutamine, 0.05 mM 2-mercaptoethanol, 1000 units/ml Leukemia Inhibitory Factor (LIF) and 10% Fetal Bovine Serum (FBS). To study the effect of G4 stabilization on origin firing, cells were grown in the presence of 0.5% DMSO or 10 μM PhenDC3 (in 0.5% DMSO). NIH3T3 cells (NIH/3T3 (ATCC CRL-1658) were grown in Dulbecco's modified Eagle's minimal (DMEM) medium supplemented with 10% FBS.

### Genetic modification using the CRISP/Cas9 technology

*Surveyor assay*. The gRNAs for targeted Cas9-driven genetic modifications were designed using the ZiFiT Targeter Software Version 4.2 (http://zifit.partners.org/ZettoniFiT/Disclaimer.aspx). The specificity of the designed gRNAs was tested in the Surveyor assay using the T7 endonuclease (ref NEB #E3321) with the primers SURV_C_S697, SURV_C_AS697 (for sequences see Supplementary Table 2). Successful modification of the chosen region was confirmed by gel electrophoresis of the obtained products (Supplementary Fig. 2B).

*Ectopic origin creation and deletion experiments*. Ectopic origin creation in mouse NIH 3T3 cells was obtained by lipofectamine (Invitrogen, ref. 18324–012) transfection of the MLM3639 plasmid expressing the Cas9 endonuclease (https://www.addgene.org/42252/), MLM3639 plasmid expressing a gRNA specific to the targeted region (gRNA insertion F, gRNA insertion R) (https://www.addgene.org/43860/), linearized pBluescript plasmid bearing the template for homologous recombination, and pBABE-puro vector encoding the puromycin resistance gene (https://www.addgene.org/34589/). Cells were selected in medium containing 2.5 μg/ml puromycin. The insertion presence was confirmed using the C3 AS1, ori1 G1, A5 S2, and ori1 C1 primers (Supplementary Table 2), and the absence of random insertions of the linearized pBluescript plasmid using the primers pBS1529S and pBS1726AS (Supplementary Table 2). Clones positive for homologous recombination were amplified for nascent strand purification.

**OGRE/G4 deletion from an endogenous origin**. Deletion of an OGRE/G4 from an endogenous replication origin was obtained by transfection of the MLM3639 plasmid expressing the Cas9 nickase (hCas9_D10A) (https://www.addgene.org/41816/), two different MLM3639 plasmids to express gRNAs specific to the targeted regions (gRNA Ori1 delG4 1 F, gRNA Ori1 delG4 1 R, gRNA Ori1 delG4 2 F, gRNA Ori1 delG4 2 R; for sequences see Supplementary Table 2), and the pBABE-puro vector encoding the puromycin resistance gene. After puromycin selection, cells were cloned and checked for the presence of mutations using the MslI restrictase that recognizes a specific sequence in the vicinity of the targeted region (for experimental outcome see Supplementary Fig. 3A). The region of interest was amplified from clones bearing mutations using the primers Ori1 742 F and Ori1 742 R (Supplementary Table 2) and subcloned in pBluescript for precise mutation mapping by sequencing.

**RNA-primed short nascent strand (SNS) DNA strand isolation**. SNS were purified as described in[4] and in Supplementary Methods. The Illumina TruSeq ChIP Sample Prep Set A (ref 15034288) was used for preparation of sequencing libraries. Samples were sequenced using the Illumina HiSeq 2000 at the MGX GenomiX facility (Montpellier). To perform local origin mapping, purified nascent strand samples were amplified by qPCR using the specific primers listed in Supplementary Table 2 with the LightCycler 480 SYBR Green Master mix (Roche, ref. 04887352001) on a LightCycler 480 II apparatus (Roche). The nascent strand enrichment was calculated as the ratio of the signal scored at origin-specific and background regions. If not otherwise specified, the statistical analysis was performed with the two-tailed, unpaired *t* test and the enrichment detected in 3 independent experiments. Differences with *p* values ≤ 0.05 were considered as statistically significant.

**Local transcription activity measurement**. Total cell RNA was extracted using the RNeasy Mini Kit (ref 74104 Qiagen) and cDNA was synthetized using the First-Strand cDNA Synthesis Kit with SuperScript II and a polyA primer (Invitrogen), according to the manufacturer's protocol. The transcription activity of selected genes was measured by qPCR with specific primers designed at the exon-intron junctions to avoid amplification from any possible DNA contaminant (Rai1c4ex3–4, Rai1 qPCR130, Gapdh ex4–5, Actb-ex2–3, Actb; see Supplementary Table 2). The relative transcription level was calculated as the transcription level found in the mutated versus parental cell line. The mean ± SD was calculated from three independent experiments and the statistical evaluation was performed with the two-tailed, paired $t$ test ($p$ value ≤ 0.05 was considered significant).

**Read mapping**. Sequenced reads were mapped against the mm10 mouse genome sequence (NCBI GRCm38) using Bowtie2. Origins identification was obtained using MACS2 (version 2.1.0, ref. [57] (narrow peaks) and SICER (broad region). MACS2 peaks overlapping SICER regions were considered as actual replication initiation sites (IS). Three biological replicates of control mouse ES cells incubated with 0.5% DMSO and two replicates of mouse ES cells incubated with 10 μM PhenDC3 were used as well as one RNase A-treated sample prior to λ exonuclease digestion (control). Only origins reproducibly present in at least two replicates in each condition were retained for further analysis. For figures representing raw data (UCSC tracks Fig. 3c, and Heatmap Fig. 3d), the mapped reads from replicates incubated with DMSO or PhenDC3 were merged for simplicity. Differential binding analysis was performed using the DESeq2 option in the DiffBind R package (version 1.12.3). The resulting $p$ values were subjected to Benjamini–Hochberg multiple testing correction to derive the false discovery rates (FDR); only sites differentially bound with a FDR ≤1% were considered as differential. As a negative control for peak clustering, correlation with chromatin marks and motif discovery, the shuffle program from the Bedtools suite (v2.25.0[29] was used to select random genomic regions of the same number and sizes as the origin peaks.

**Genomic localization**. Origin localized at promoters (2 kb upstream TSS) in transcribed and intergenic regions were identified using the GenomicRanges R package and the TxDb.Mmusculus.UCSC.mm10.knownGene, version 3.0.0, genome database. For negative controls, the IS coordinates were shuffled 1000 times while keeping the chromosomal distribution of each class and avoiding long regions lacking genomic information.

**G4 assignation**. Putative G4 were identified using the G4-Hunter algorithm[26] and a score higher than 2. An IS was considered as G4-positive if the G4 (with a G4Hunter_score ≥2) was located ±500 bp from its center. The G4-Hunter score evaluates the propensity of a sequence to form a G4. A sequence with a G4-Hunter score higher than 2 should form a G4; to date, no sequence with such score was unable to form a G4 in classical experimental conditions (37 °C, neutral pH, 100 mM NaCl or KCl).

**G4 profile**. G4 location profiles were computed by counting the "G4 location" at the base pair level at ±1 kb from the IS for each origin class. Then, the sum of the coverage, or the G4 ratio for each group, was computed to obtain the G4 profiles for each origin class. Profiles of G4 on the minus strand (CCC) were oriented on the (+) strand.

**RNA-seq and differential gene expression**. Total RNA was extracted using the RNeasy Mini Kit (Qiagen; cat 74104), and libraries were prepared using the Illumina TruSeq Stranded mRNA Sample Preparation Kit and sequenced using an Illumina HiSeq 2500 apparatus at the MGX GenomiX facility (Montpellier). The TopHat software (version 2.1.1) was used for splice junction mapping with Bowtie2 (version 2.2.9) for mapping reads. Reads counting on genes was done using HTSeq-count (version 0.6.1p1). Data were normalized to the relative log expression implemented in edgeR (version 3.16.5), and the statistical analysis to identify differentially expressed genes was performed using DESeq2 (version 1.14.1). Differential gene expression was considered when the adjusted $p$ value was ≤0.05 after multi-testing correction using the Benjamini–Hochberg method.

**Genomic Set Enrichment Analysis (GSEA)**. The GSEA was performed using the R package fgsea (version 1.2.1) and the data obtained in the differential RNA-seq analysis. Genes were ranked from upregulated to downregulated using the adjusted $p$ value and the sign of the fold change obtained from the DESeq2 analysis. The enrichment set test was computed with the genes associated with one of the origin classes (suppressed, new, etc.), and the $p$ value was computed using 10,000 permutations (origin-gene associations).

**De novo motif discovery**. The RSAT peak-motifs program[58] was used to detect de novo motifs around the IS summits from −1 kb to +1 kb. Among the results, the motif found by positions-analysis for 6–7 nt and with the lowest e-value and the highest significance was selected.

**Episomal DNA replication assay**. The HEK-293 cell line that stably expressing EBNA1 (HEK293 EBNA1+) was cultured in DMEM with 10% fetal calf serum and 220 μg/ml neomycin. The HEK293 cells was originally received from DSMZ (DSMZ No: ACC 305). CMV-EBNA1 was stably integrated into the chromosome after linearization and selected with 220 μg/ml Neomycin. Episomal replication was assayed using the Dpn1 digestion method[59]. The reporter plasmids (2 μg) containing the various origin variants were transfected in HEK293 cells that express EBNA1, and the transfection efficiencies were verified by visualizing GFP-positive cells. Six days post-transfection, cells were harvested using the protocol described by Hirt et al.[60] Isolated DNA was purified by phenol-chloroform extraction and digested with 40 U DpnI (NEB) in the presence of RNase (Roche). Digested DNA (300 ng) was electroporated in Electromax DH10B competent cells (Invitrogen) and ampicillin-resistant colonies, representing the number of recovered plasmids, were counted to estimate the replication efficiency of the episome (presented as the mean ± standard deviation calculated from 3 to 7 independent experiments). Statistical significance was evaluated using the two-tailed, unpaired $t$ test ($p$ value ≤ 0.05 was considered as significant).

**X. laevis egg extract and DNA replication kinetics**. Low Speed Egg (LSE) and High-Speed Egg extracts (HSE) were prepared as previously described[61,62] (for details see Supplementary Methods). Chromosomal DNA replication was assayed by adding demembranated X. laevis sperm nuclei to extracts supplemented with [α-32P]-dCTP. For competition assays, extracts were incubated with 2 ng/μl of oligonucleotides (or shared salmon sperm as control, or ultrapure water) at 22 °C for 5 min before sperm nuclei addition. DNA synthesis was monitored by TCA precipitation. Incorporated acid-insoluble material was spotted onto Whatman glass microfiber filters, grade GF/C, and then precipitated with 5% TCA solution containing 2% pyrophosphate. After ethanol washes, filters were dried and the incorporated TCA-precipitated radioactivity was counted in scintillation liquid. M13 replication kinetics were assessed using 400 ng of ssDNA per 50 μl of HSE[62] pre-incubated or not with oligonucleotides Sperm chromatin purification for protein-binding monitoring was performed as previously described[61] Briefly, chromatin pellets were resuspended in 2× LB (0.125 M Tris-HCl pH 6.8, 4% SDS, 20% glycerol, 10% 2-β-¬mercaptoethanol and 0.004% bromophenol blue), denatured at 95 °C for 5 min, and then stored at −20 °C or immediately analyzed by SDS-PAGE, using gradient Bis-Tris gels (Thermo Fisher Scientific).

**Antibodies**. The antibodies used in this work were against: H3 (Abcam, ab1791, dilution 1/2000), H2B (Abcam, ab1790, dilution 1/2000), phosphorylated CHK1 (Cell Signaling, 2341 S, dilution 1/250), PCNA (Sigma, P8825, dilution 1/2500), RPA34[62] (dilution 1/500), MCM3[61] (dilution 1/2000), CDC45[63] (dilution 1/1000), ELYS[31,64] (dilution 1/500), MCM4[63] (dilution 1/1000), anti-Chk1 (dilution 1/500), anti-ORC5 (dilution 1/1000), anti-CDC6[63] (dilution 1/500), OCT4 (Abcam, ab19857, dilution 1/500), actin (Sigma, A4700, dilution 1/500), HRP-linked ECL anti-mouse IgG (GE Healthcare, NA931V, dilution 1/4000), HRP-linked ECL anti-rabbit IgG (GE Healthcare, NA934V, dilution 1/4000) (For details see Supplementary Table 7).

**Spectroscopic studies**. Isothermal difference spectra (IDS) and circular dichroism (CD) measurements were performed as previously described[17,65]. Briefly, the sequences were tested at 4 μM strand concentration in 10 mM LiCaco pH 7.2 with 100 mM KCl. IDS were obtained by computing the difference between the absorbance spectra of unfolded and folded oligonucleotides that were recorded before and after addition of 100 mM KCl, respectively, at 25 °C. CD spectra were recorded at 20 °C after IDS (in K+) on a JASCO-1500 spectropolarimeter using 1 cm path length quartz cuvettes.

**FRET melting assay and FRET competition assay**. The tested G4 sequences (Table S3) were labeled with Fam on 5′ and Tamra on 3′. Each sequence was pre-folded at 0.2 μM in 10 mM LiCaco pH 7.2 supplemented with 10 mM KCl and 90 mM LiCl before adding the PhenDC3 ligand (1 μM). Stabilization (increase in $T_{1/2}$, expressed in °C) was plotted for each G4-forming sequence; as a control a dsDNA of the same length were used. In the FRET competition assay, stabilization ($\Delta T_{1/2}$, in °C) of the human telomeric quadruplex F21T by 0.5 μM PhenDC3 was analyzed in the presence/absence of increasing amounts of each G-rich origin sequence (3 or 10 μM strand concentration).

**Reporting summary**. Further information on research design is available in the Nature Research Reporting Summary linked to this article.

## Data availability

The SNS-seq and RNA-seq data are deposited at the NCBI GEO (GSE126477) [https://www.ncbi.nlm.nih.gov/geo/query/acc.cgi?acc=GSE126477] and [http://rsat-tagc.univ-mrs.fr/g4/g4_data.html]. R scripts used for figure creation are deposited under [https://github.com/LacroixLaurent/G4Hunter_mm10_Ori] and http://rsat-tagc.univ-mrs.fr/g4/g4_data.html]. Data supporting the findings of this study are available within the paper and its supplementary information files, including uncropped scans of the most important blots. All the data are available from the authors upon reasonable request.

## Code availability

For MACS2 see https://github.com/taoliu/MACS. For SICER see https://home.gwu.edu/~wpeng/Software.htm. For GenomicRanges: https://bioconductor.org/packages/release/bioc/html/GenomicRanges.html. For DESeq2 see http://bioconductor.org/packages/release/bioc/html/DESeq2.html For G4-Hunter see https://github.com/LacroixLaurent/G4HunterPaperGit. For Fgsea https://bioconductor.org/packages/release/bioc/html/fgsea.html. For RSAT http://rsat-tagc.univ-mrs.fr/rsat/RSAT_home. A reporting summary for this Article is available as a Supplementary Information file.

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

## Acknowledgements

We would like to thank Jacques van Helden (Aix-Marseille University) and our lab members for helpful discussions. We thank Marie-Paule Teulade-Fichou (Institut Curie, Orsay, France) for providing us PhenDC3. We thank J. Walter for the anti-CDC45 antibodies, I. Mattaj and J. Blow for the anti-ELYS antibody. We are grateful to the Genotoul Bioinformatics Platform Toulouse Midi-Pyrenees for computing and storage resources. We also thank E. Andermarcher for critical reading of the manuscript. The research leading to these results has received funding from the European Research Council (FP7/2007–2013 Grant Agreement no.233339). This work was also supported by the ARC foundation and ANR14-CE10-0019, and by the MSDAVENIR Fund GENE-IGH. PP was supported by a post-doctoral fellowship from the ARC Foundation (Fondation ARC pour la Recherche sur le Cancer).

## Author contributions

M.A., P.C. and A.A. contributed equally. M.M. proposed the project and the experimental system, P.P. designed and performed the majority of experiments, P.P. and P.C. designed the genome-modification experiments, L.L., M.A. and B.B. performed the bioinformatics analysis, A.A. performed the experiment with X. laevis egg extacts, I.P. cultured the cells and did FACS analysis, A.G. and J.L.M. designed and analysed in vitro G4-formation assays, J.D. and A.S. performed episome replication assay, P.P. and M.M. wrote and revised the manuscript. All the authors read and approved the final manuscript.
