## [Peer Review File · Nature Communications]

Reviewers' comments:

Reviewer #1 (Remarks to the Author):

In this manuscript authors address the relevance of genetic and genomic features on the activity of DNA replication origins. They study in detail the requirement of OGRE, which can potentially form G4 structures, in origin function. First they demonstrate that several OGRE actually form G4 structures. Then they focus on one origin and show that removal of the OGRE/G4 sequence affects origin activity whereas insertion of that element into a genomic site devoid of origin activity leads to ectopic origin activity. The use of PhenDC3, a compound that stabilizes G4 structures, shows little effect of origin activity. Using a combination of molecular and genomic strategies they identify five classes of origin according to their response to PhenDC3. Finally, they address the question of whether G4 are relevant for association of pre-RC and further replication factors using an in vitro *Xenopus* system and conclude that OGRE/G4 appear to be relevant for activation of DNA replication but not for pre-RC formation. This is a topic of general interest and the experimental design is adequate. The experiments have been also carefully carried out with appropriate controls and analyses. I have a few comments mainly dealing with points that I consider not sufficiently treated or that need further clarification.

1. A more detailed discussion of the relationship between the localization of OGRE/G4 elements and the localization of pre-RC should be enlightening, in particular because the authors address this question experimentally.

2. Page 4, first sentence, first section of Results. This is a general statement that is missing that studies of origin activity using SNS have been also carried in *Arabidopsis* (Costas et al., 2011). Otherwise, it should be specified that the list of references refer exclusively to metazoan systems. Also, a comment on the different sequence features of the material used in the CD and IDS experiments should serve to clarify the main point.

3. Page 5. Line 1. According to Fig. S2A there are two clear regions with high amount of NS reads with a valley in between. The OGRE/G4 is located 280 nt upstream of the IS. How is the IS defined? The midpoint of the entire region? Of one of them? How this sub-region is chosen and why?

4. Page 5, line 3. What is the criterion used to choose a piece of 1907 bp for this experiment? Is this the minimum size that confers ectopic origin activity? I wonder if further experiments have been done (or can be done) to address this more specifically.

5. Page 5, fig. 1D. The antisense OGRE/G4 appears to have a significant effect. Please comment.

6. Page 7, line 1. Origins belonging to the insensitive class have much lower amount of NS reads than the reduced or suppressed classes (Fig. 3E). Conversely, the reduced class origins are particularly strong origins whereas origins in the new class that appears after G4 stabilization are not. Is all this telling something about the nature of these origins and/or about their response to the drug?

7. Page 7, a few lines below. As indicated, most origins (77.9%) are insensitive to G4 stabilization and two possible explanations are given. I wonder if other possibilities should be considered, e.g., these (and possibly other) origins function depends of its local landscape (sequence, chromatin, transcription). This and other alternatives should be discussed.

8. Page 12. It is suggested that in the in vitro system oligonucleotides do not interfere with licensing but rather the switch to the elongation complex. Do the authors mean that (some) replication factors are sequestered by adding the oligonucleotides? I wonder if this could this be reverted by adding back purified components to the reaction shown in 6A? A more detail model

/hypothesis explaining the effect of G4 could serve to clarify this point.

9. Authors may consider to make more explicit statements/discussion highlighting the results obtained here compared to previous studies on the role of G4.

Reviewer #2 (Remarks to the Author):

Porok et al. Involvement of G-quadruplex regions in mammalian replication origin activity

Telomeres, centromeres and replication origins are the functional elements required for proper replication, segregation and stability of linear eukaryotic chromosomes. While telomeres and centromeres are structurally and functionally well defined, replication origins still represent the „holy grail“ of chromosome research. In contrast to bacteria and yeast the replication origins of higher eukaryotes are not defined by a specific DNA sequence but their activation seems to depend on several epigenetic factors, f.e. an interplay between transcription and replication or a specific chromatin structure. Moreover, the genome contains many more potential replication origins than are actually used and the mechanisms how an origin is selected to be active is largely unknown. Several years ago a genome wide study revealed that many active replication origins are associated with a GC rich sequence having the potential to form a G-quadruplex structure (G4). This structure was regarded for many years as a structural curiosity only formed *in vitro*, but meanwhile there is very good evidence that it occurs *in vivo* and the formation of this structure has been correlated with telomere metabolism, transcription, replication and translational control (a more recent review on this structure would be useful to have in the reference list). Therefore, the observation that a potential G-quadruplex forming sequence is associated with many replication origins was regarded as a real breakthrough and indeed some evidence was provided that this sequence might be involved in the initiation of replication. The present manuscript sets out to understand the function of this sequence in replication activation in more detail and uses for this study well defined genomic replication origins as well as an episomal replicon based on EBV.

The main results presented in this study are 1. That the deletion of the G4 motif (OGRE) reduces origin activity in those origins normally associated with this motif. 2. Insertion of this motif leads to the formation of a new origin, 3. An origin associated with an OGRE can replace the EBV origin and deletion of the OGRE reduces replication efficiency, 4. By using the G4 ligand PhenDC3 origins with different properties could be identified and finally 5. That G4 forming sequences in *Xenopus* eggs act as competitor sequences and affect replication.

This is a very complex study and nothing is wrong with the experimental design or their interpretations. But for my feeling authors asked too many questions with the result that none of them is answered satisfactorily. As far as I can see we have the following facts: 1. 60% of origins are associated with an OGRE and these sequences are important for replication initiation. 2. Origins in promoter regions are functionally not dependent on that sequence.

These facts imply that 1. the OGREs and ongoing transcription both must create an environment that allows replication initiation. The most likely environmental change induced by both is chromatin structure. This question could be easily addressed in the episomal context and 2. If a G-quadruplex structure is required for replication initiation its formation must be tightly regulated. The experiments showing that G4s are competitors and affect replication activity in *Xenopus* eggs clearly show that some components specifically bind to them and probably regulate the formation of G quadruplex formation as well as their resolving. Of course, the identification of such components is not trivial but in the end it will be important to fully understand the function of the OGREs and the regulation of replication initiation and the selection of replication origins in eukaryotic chromosomes. Perhaps a first step towards such an understanding would be to use not only a G4 binding ligand but also drugs either inducing or inhibiting G4 formation.

In summary: this is a very nice paper presenting important informations but unfortunately these informations do not really contribute to our understanding of replication control in higher eukaryotes. Perhaps concentrating on fewer questions could improve this work significantly.

Minor Comments

p.5 „Insertion of a 500bp mouse OGRE/G4 containing origin at the place of OriP...“ What happens if this element is inserted at other sites of the plasmid?

The manuscript is not easy to read and some sentences are hardly or not understandable. One example is on p.9 „Analysis of the genomic location in control cells and...“. I do not understand.

Reviewer #3 (Remarks to the Author):

In this manuscript the authors present convincing data arguing that G-quadruplex forming motives have a strong impact on the landscape of replication origins. Elegant genomic (and plasmid) approaches are used to show causality of the motifs in establishing new origins, which is further supported by genome-wide mapping of origins in (mouse) cells exposed to G-quadruplex stabilising compound PhenDC3. The in-depth analysis (making different subcategories of affected or not-affected origins) and the additional in vitro work nicely and convincingly strengthens the overall concept, which is important to both the field and to a broader community.

I also think that the paper is very clear and excellently written. It was a joy to read.

I only have one smaller point: I feel that the paper will be important for our thinking and by itself provides sufficient novelty. Nevertheless, the publication from the Prioleau lab (Valton et al., 2104) is very relevant and constitutes important original work. I feel that the authors do not give enough credit by using the phrasing "Some function evidence....". Given the overlap in modelbuilding I would strongly suggest to refer to that paper in a more properly fashion.

Reviewer 1:

In this manuscript authors address the relevance of genetic and genomic features on the activity of DNA replication origins. They study in detail the requirement of OGRE, which can potentially form G4 structures, in origin function. First they demonstrate that several OGRE actually form G4 structures. Then they focus on one origin and show that removal of the OGRE/G4 sequence affects origin activity whereas insertion of that element into a genomic site devoid of origin activity leads to ectopic origin activity. The use of PhenDC3, a compound that stabilizes G4 structures, shows little effect of origin activity. Using a combination of molecular and genomic strategies they identify five classes of origin according to their response to PhenDC3. Finally, they address the question of whether G4 are relevant for association of pre-RC and further replication factors using an in vitro *Xenopus* system and conclude that OGRE/G4 appear to be relevant for activation of DNA replication but not for pre-RC formation. This is a topic of general interest and the experimental design is adequate. The experiments have been also carefully carried out with appropriate controls and analyses. I have a few comments mainly dealing with points that I consider not sufficiently treated or that need further clarification.

Reviewer 1 found the topic addressed “of general interest and the experimental design is adequate. “. He (she) also found “The experiments have been also carefully carried out with appropriate controls and analyses.” Our answers to his (her) comments are as below:

Comment 1

A more detailed discussion of the relationship between the localization of OGRE/G4 elements and the localization of pre-RC should be enlightening, in particular because the authors address this question experimentally.

We thank the reviewer for this question as this will certainly provide the opportunity to clarify our findings. In the revised version of our manuscript we now present a model for a potential role of OGRE/G4 element in the replication initiation. We stress that a replication origin is generally composed of two main elements, from *E. coli* to human replication origins: the DNA site where the replication origin complex assemble, and the initiation site of DNA synthesis, downstream of it. We already reported that the OGRE/G4 element is located about 250 bp upstream of the initiation site. This was confirmed in chicken cells (Valton et al., EMBO J. 2014, 33(7), 732-46, cited). G4-structures have been shown to be recognized by crucial players in DNA replication initiation, including ORC, MTBP proteins, and Rif1. Moreover, this position is nucleosome-free, therefore facilitation the assembly of the replication complex. Based on these and our observation, the use of G4 elements for origin recognition by factors regulating the binding or activation of preRCs is a reasonable hypothesis. This is now discussed in the revised manuscript, page 3 and 17 with the corresponding references and illustrated by the model presented on the Figure 7.

Comment 2

Page 4, first sentence, first section of Results. This is a general statement that is missing that studies of origin activity using SNS have been also carried in Arabidopsis (Costas et al., 2011). Otherwise, it should be specified that the list of references refer exclusively to metazoan systems.

In the first version of manuscript, we effectively stated that we referred to the metazoan systems. However, we agree that the SNS method has also been successfully used for several studies in Arabidopsis, and we comment it now, page 4 of the revised manuscript

Also, a comment on the different sequence features of the material used in the CD and IDS experiments should serve to clarify the main point

To test their propensity to form a G4-structure, we selected different origins in different chromatin domains, transcription status and replication activity. Because each sequence needed to be synthesized and tested by CD and IDS, we did a selection of 7 G4-forming sequences associated with insensitive origin class and 9 sequences associated with new origin class. The bioinformatical prediction for a potential of G4-structure was first tested at the bioinformatical level, using the G4H software (similar results were obtained with the Quadparser software), and indicated a high capacity for G4-formation for all tested sequenced. Experimental analysis of G4 formation by CD and IS confirmed the bioinformatic indications. Clarifications have been done in the revised Supplemental Table S1.

We are also now supplying the experimental analyses of the wt and mutated sequences of Ori1 and Ori2 by circular dichroism and isothermal differential spectra. Noteworthy, they strongly confirm the bioinformatics predictions of G4-forming potential. These data are now presented in the Supplemental Figure S3B and S3C and commented on pages 6 and 7 of the revised manuscript.

Comment 3.

Page 5. Line 1. According to Fig. S2A there are two clear regions with high amount of NS reads with a valley in between. The OGRE/G4 is located 280 nt upstream of the IS. How is the IS defined? The midpoint of the entire region? Of one of them? How this sub-region is chosen and why?

The replication origin positions were defined in a genome-wide manner using MACS2 and SICER peaks calling softwares, as previously described (Cayrou et al., Genome Res. 2015, 25(12), 1873-852015, cited). Macs2 Narrow Peaks overlapping with Sicer broader regions were scored as the replication initiation sites. MACS2 output permit to precise in an individual origin the location with the highest enrichment score, referred as the summit which was considered as the replication origin initiation site (IS). Therefore, the origin initiation site is the highest NS-enrichment score over the initiation region. We are now adding these precisions in our revised manuscript, page 5. The revised manuscript also contains a modified Figure S2A with the summit location that is positioned 240 nt downstream to the G4-forming sequence for this origin.

Comment 4

Page 5, line 3. What is the criterion used to choose a piece of 1907 bp for this experiment? Is this the minimum size that confers ectopic origin activity? I wonder if further experiments have been done (or can be done) to address this more specifically.

The selection of a fragment for the ectopic insertion was based on all our duplicates of the genome-wide replication origin mouse ES cell maps available in our laboratory (5 independent duplicates). We selected a strong and highly reproducible replication origin and found that it confers the replication activity into the ectopic position. The ectopic insertion experiments were experimentally quite challenging, and to perform more analysis aimed to target the G4, we performed deletions of the G4-forming sequence in an endogenous replication origin, which resulted in a strong decrease of the replication origin activity. It is possible that other elements close to the G4 also help to obtain an optimal activity, or play the role of auxiliary elements. However, our experiment performed both by ectopic insertion and mutation analysis as well as our data using plasmid DNAs, clearly agree with a main role of OGRE/G4 element in replication origin activity.

Comment 5

Page 5 Fig. 1D. The antisense OGRE/G4 appears to have a significant effect. Please comment.

As reported before (see comment 1 answer), the OGRE/G4 presence is orientated relative to the initiation site, as initiation occurs always downstream to the OGRE/G4. So, when the antisense sequence is used, initiation should also occur, but in the initiation site will be in the other direction. This is illustrated in the figure below which has been also added to the manuscript (new Figure 7) in order to comment both this comment and comment 1.

The efficiency can possibly vary according to the genetic and epigenetic properties found in the reverse direction. Replication efficiency of the episome using the sense and antisense OGRE/G4 differ slightly but not significantly. In the revised version of manuscript, we are clarifying this point now (page 6).

Comment 6

Page 7, line 1. Origins belonging to the insensitive class have much lower amount of NS reads than the reduced or suppressed classes (Fig. 3E). Conversely, the reduced class origins are particularly strong origins whereas origins in the new class that appears after G4 stabilization are not. Is all this telling something about the nature of these origins and/or about their response to the drug?

This is an interesting point that we should have commented in the first version of the manuscript. The replication efficiency scored for origins from insensitive, new, reinforced and suppressed class ranged between 100 and 200 reads per origin and was slightly modified upon G4-stabilization. Only origins from the reduced class exhibited stronger replication activity before G4-stabilisation. These origins were quite strong because of the presence of both G4 and active transcription. This is shown later in the manuscript. The decreased transcription activity at these origins upon G4 stabilisation decreased the stimulating effect of transcription of these origins. We also think that the formation of 4708 new replication origins resulted in a more homogenous read distribution between active origins with less need of particularly strong origins. This comment is now included in the revised manuscript page 10.

Comment 7

Page 7, a few lines below. As indicated, most origins (77.9%) are insensitive to G4 stabilization and two possible explanations are given. I wonder if other possibilities should be considered, e.g., these (and possibly other) origins function depends of its local landscape (sequence, chromatin, transcription). This and other alternatives should be discussed.

We found that most origins have an OGRE, and that their corresponding G4 have a high G4Hunter score, so they are really prone to G4 formation, and they should not need a ligand to be active. We agree that G4 are not the only important feature for replication origin activity. The chromatin landscape and transcription have been already shown to affect DNA replication origin activity, and we provide later in the manuscript some new evidence for this. We nevertheless added a sentence in this paragraph in agreement with the reviewer comment (page 8).

Comment 8, Page 12

It is suggested that in the in vitro system oligonucleotides do not interfere with licensing but rather the switch to the elongation complex. Do the authors mean that (some) replication factors are sequestered by adding the oligonucleotides? I wonder if this could be reverted by adding back purified components to the reaction shown in 6A? A more detail model /hypothesis explaining the effect of G4 could serve to clarify this point.

Using *Xenopus laevis* egg extracts we observed a strong inhibition of DNA replication in the presence of G4-oligonucleotides but not with random or AT-rich oligonucleotides. The

analysis of chromatin recruitment of replication licencing and activating factors revealed a delay in the binding of the firing-associated proteins. DNA replication in *X. laevis* egg extracts needs highly concentrated egg extracts. Moreover, the complementation analysis of the extract suggested by the reviewer would need first to purify several components of the replication initiation complex and also to get them in a very concentrated form, in order to avoid changing the final reaction volume. At the moment, this would be technically highly challenging. However, we clarified the principle of oligonucleotide competition assays in *Xenopus* egg extracts in the corresponding paragraph, page 11, and in page 16 of the revised manuscript.

Comment 9

Authors may consider to make more explicit statements/discussion highlighting the results obtained here compared to previous studies on the role of G4.

We believe our study is the first to functionally address the importance of G4 formation at several functional levels. This includes (i) in-vivo analyses in a genome-wide context, the relation to genome organisation, chromatin accessibility, epigenetic landscape; (ii), in a plasmid context; by mutation analyses done strictly on endogenous sequences and not on artificial constructions; (iii), in a well- established in-vitro replication system. We tried to make more explicit comments in consideration of previous studies on the G4 in the discussion paragraph of the revised manuscript.

Reviewer 2

Porok et al. Involvement of G-quadruplex regions in mammalian replication origin activity

Telomeres, centromeres and replication origins are the functional elements required for proper replication, segregation and stability of linear eukaryotic chromosomes. While telomeres and centromeres are structurally and functionally well defined, replication origins still represent the „holy grail“ of chromosome research. In contrast to bacteria and yeast the replication origins of higher eukaryotes are not defined by a specific DNA sequence but their activation seems to depend on several epigenetic factors, f.e. an interplay between transcription and replication or a specific chromatin structure. Moreover, the genome contains many more potential replication origins than are actually used and the mechanisms how an origin is selected to be active is largely unknown. Several years ago a genome wide study revealed that many active replication origins are associated with a GC rich sequence having the potential to form a G-quadruplex structure (G4). This structure was regarded for many years as a structural curiosity only formed in vitro, but meanwhile there is very good evidence that it occurs in vivo and the formation of this structure has been correlated with telomere metabolism, transcription, replication and translational control (a more recent review on this structure would be useful to have in the reference list). Therefore, the

observation that a potential G-quadruplex forming sequence is associated with many replication origins was regarded as a real breakthrough and indeed some evidence was provided that this sequence might be involved in the initiation of replication. The present manuscript sets out to understand the function of this sequence in replication activation in more detail and uses for this study well defined genomic replication origins as well as an episomal replicon based on EBV.

The main results presented in this study are 1. That the deletion of the G4 motif (OGRE) reduces origin activity in those origins normally associated with this motif. 2. Insertion of this motif leads to the formation of a new origin, 3. An origin associated with an OGRE can replace the EBV origin and deletion of the OGRE reduces replication efficiency, 4. By using the G4 ligand PhenDC3 origins with different properties could be identified and finally 5. That G4 forming sequences in *Xenopus* eggs act as competitor sequences and affect replication.

This is a very complex study and nothing is wrong with the experimental design or their interpretations. But for my feeling authors asked too many questions with the result that none of them is answered satisfactorily. As far as I can see we have the following facts: 1. 60% of origins are associated with an OGRE and these sequences are important for replication initiation. 2. Origins in promoter regions are functionally not dependent on that sequence.

These facts imply that 1. the OGREs and ongoing transcription both must create an environment that allows replication initiation. The most likely environmental change induced by both is chromatin structure. This question could be easily addressed in the episomal context and 2. If a G-quadruplex structure is required for replication initiation its formation must be tightly regulated. The experiments showing that G4s are competitors and affect replication activity in *Xenopus* eggs clearly show that some components specifically bind to them and probably regulate the formation of G quadruplex formation as well as their resolving. Of course, the identification of such components is not trivial but in the end it will be important to fully understand the function of the OGREs and the regulation of replication initiation and the selection of replication origins in eukaryotic chromosomes. Perhaps a first step towards such an understanding would be to use not only a G4 binding ligand but also drugs either inducing or inhibiting G4 formation.

In summary: this is a very nice paper presenting important informations but unfortunately these informations do not really contribute to our understanding of replication control in higher eukaryotes. Perhaps concentrating on fewer questions could improve this work significantly.

Major comment

The reviewer states that “This is a very complex study and nothing is wrong with the experimental design or their interpretations”. He (she) also feel that we have “ asked too many questions”, but also think that “In summary: this is a very nice paper presenting important informations”.

Our aim was indeed to use several *independent* experimental strategies to ask whether the significance of G4 for initiation of DNA replication could be established by different approaches. We believe that such thorough functional analysis on G4 and initiation of DNA replication is not yet not available at present.

The reviewer also propose that a *first step towards such an understanding would be to use not only a G4 binding ligand but also drugs either inducing or inhibiting G4 formation.*

The principal action of G4 stabilizing or destabilising molecules is to displace the equilibrium between the folded and unfolded G4-form. Any molecule that recognizes and bind to a folded G4 increases the thermodynamic stability and the half-life of this form. Actually, there is no universal G4-destabilising molecule that could be used as a general G4-destabilizer. The recently reported use of sequence-specific oligonucleotides for G4-unfolding (Rouleau et al., Nucleic Acids Res. 2015, 43, 595-606) unfortunately cannot be applied in a genome-wide approach. In our study the site-specific G4-suppression was obtained using CRISPR-Cas9 driven deletion of a chosen G4 that allowed to stably prevent the formation of this structure. Eventually, the destabilization of a given structure might be obtained by the stabilization of another alternative structure, which would be double or single-stranded DNA stabilisation. However, such manipulation would unavoidably prevent the formation of other secondary form of DNA or RNA and therefore would have less specificity. A global G4-destabilization would also lead to deleterious effect for cells because G4 are important for the regulation of telomeres.

Minor Comments

p.5, Insertion of a 500 bp mouse OGRE/G4 containing origin at the place of OriP...“ What happens if this element is inserted at other sites of the plasmid?

The EBV system has a long-standing history to serve as genetic tool to study replication initiation. OriP of EBV consists of two functional, EBNA1 dependent elements. The family of repeats (FR) mediates plasmid segregation by piggy-backing the episome to host chromosomes (Marechal V et al., J Virol. 1999, 73, 4385-4392). FR consists of 20 repeats of EBNA1 binding sites; at least 7 intact binding sites are required for efficient segregation. Single cell studies revealed that this process occurs with an efficacy of 88% (Nanbo A. et al., EMBO J 2007, 26, 4252-4262). The dyad symmetry element consists of two pairs of EBNA1 binding dyads. Importantly as well, replication occurs ORC dependently once per cell cycle in synchrony with chromosome replication with an efficacy of 84% (Nanbo A. et al., EMBO J 2007, 26, 4252-4262; Chaudhuri B. et al., Proc Natl Acad Sci USA 2001, 98, 10085-10089; Dhar S.K. et al., Cell 2001, 106, 287-296; Schepers A. et al., EMBO J 2001, 20, 4588-4602).

This has been now also precised page 5 and 6 of the revised manuscript. As deletion of DS renders oriP-plasmids replication incompetent, other genetic elements can be introduced to study their ability to support DNA replication. Very early studies by the M. Calos indicated that these elements could also be introduced at other sites than DS and support DNA replication (Krysan PJ and Calos MP Proc Natl Acad Sci USA 1991, 11, 1464-1472; Krysan PJ et al., Mol Cell Biol 1993, 13, 2688-2896; Krysan PJ et al, Mol Biol Cell. 1989, 9, 1026-1033). These studies have been the first using EBV-based vectors as autonomous replicating vector and introduced foreign DNA at another site than DS (Krysan PJ and Calos MP Proc Natl Acad Sci USA 1991, 11, 1464-1472; Krysan PJ et al., Mol Cell Biol 1993, 13, 2688-2896).

The manuscript is not easy to read and some sentences are hardly or not understandable. One example is on p.9, Analysis of the genomic location in control cells and....". I do not understand.

We checked in the Nature Com PDF of our manuscript and the wording is genomic location and not genomic location. We believe that there was a problem with the transmission of the pdf. We also noticed that the sentence was not grammatically correct, and we revised this part accordingly, page 10.

Reviewer 3

In this manuscript the authors present convincing data arguing that G-quadruplex forming motives have a strong impact on the landscape of replication origins. Elegant genomic (and plasmid) approaches are used to show causality of the motifs in establishing new origins, which is further supported by genome-wide mapping of origins in (mouse) cells exposed to G-quadruplex stabilising compound PhenDC3. The in-depth analysis (making different subcategories of affected or not-affected origins) and the additional in vitro work nicely and convincingly strengthens the overall concept, which is important to both the field and to a broader community.

I also think that the paper is very clear and excellently written. It was a joy to read.

We thank the reviewer for his positive appreciation and enthusiasm of our paper.

Minor point:

I feel that the paper will be important for our thinking and by itself provides sufficient novelty. Nevertheless, the publication from the Prioleau lab (Valton et al., 2014, cited) is very relevant and constitutes important original work. I feel that the authors do not give enough credit by using the phrasing "Some function evidence....". Given the overlap in modelbuilding I would strongly suggest to refer to that paper in a more properly fashion.

We somewhat agree, although we had already precised "Some functional evidence for the use of this element was reported in chicken cells in a 1.1kb fragment of the β -globin replication origin flanked by an HS4 insulator included close to a blasticidin resistance

transgene under the control of the strong actin promoter (Valton et al., EMBO J. 2014, cited). However, it is unclear whether this result can be translated to other model systems, and no analysis has been done so far on a natural replication origin, at its original site or at an ectopic position.” We agree that “Some functional evidence...” could be misinterpreted, so it now reads “A functional evidence...” (page 3).

Historically, the first evidence for a repeated G-rich element (OGRE) present at replication origin was in Cayrou et al, Genome Research 2011 (cited), and then reported to be G4 potential in Cayrou et al, Cell Cycle 2012 (cited). Then, a first functional evidence that corroborated the use of OGRE/G4 elements in replication origin activity was indeed reported in chicken cells, as mentioned above. However, it was for a quite artificial construction flanked by an insulator and under the control of a strong transcriptional promoter. Our analyses were strictly performed on endogenous sequences, either in an ectopic position or at their endogenous place. We believed it was also fair to mention the exactly experimental strategy used. Please also note that as for the model building for the position of the OGRE/G4 element (Figure 9 of Valton et al, cited), it is the same than our proposed model in Figures 2 and 3 of our previous paper defining OGRE/G4 elements (Cayrou et al, Cell Cycle 2012 which was following Cayrou et al, Genome Research 2011). In the current manuscript, we are providing an experimental support for the functional involvement of G4-structure in replication origin specification.

REVIEWERS' COMMENTS:

Reviewer #1 (Remarks to the Author):

The authors have made the changes, as suggested in my report, including new comments and clarifying the points. Also they have argued satisfactorily the aspects that were raised. Therefore, I am satisfied with the revised version.

Reviewer #2 (Remarks to the Author):

The authors address all the concerns raised by all reviewers. As I wrote before this is a nice report but fortunately still leaves space for further studies on the regulation and function of G4 in replication origins. I am quite happy with that version. Hans J. Lipps